# On the Convergence of Gradient Flow on Multi-layer Linear Models

## Abstract

In this paper, we analyze the convergence of gradient flow on a multi-layer linear model with a loss function of the form $f(W_1 W_2 \cdots W_L)$. We show that when $f$ satisfies the gradient dominance property, proper weight initialization leads to exponential convergence of the gradient flow to a global minimum of the loss. Moreover, the convergence rate depends on two trajectory-specific quantities that are controlled by the weight initialization: the *imbalance matrices*, which measure the difference between the weights of adjacent layers, and the least singular value of the *weight product $W = W_1 W_2 \cdots W_L$*. Our analysis provides improved rate bounds for several multi-layer network models studied in the literature, leading to novel characterizations of the effect of weight imbalance on the rate of convergence. Our results apply to most regression losses and extend to classification ones.

## 1 Introduction

The mysterious ability of gradient-based optimization algorithms to solve the non-convex neural network training problem is one of the many unexplained puzzles behind the success of deep learning in various applications (Krizhevsky et al., 2012; Hinton et al., 2012; Silver et al., 2016). A vast body of work has tried to theoretically understand this phenomenon by analyzing either the loss landscape or the dynamics of the training parameters.

The *landscape-based analysis* is motivated by the empirical observation that deep neural networks used in practice often have a benign landscape (Li et al., 2018a), which can facilitate convergence. Existing theoretical analysis (Lee et al., 2016; Sun et al., 2015; Jin et al., 2017) shows that gradient descent converges when the loss function satisfies the following properties: 1) all of its local minimums are global minima; and 2) every saddle point has a Hessian with at least one strict negative eigenvalue. Prior work suggests that the matrix factorization model (Ge et al., 2017), shallow networks (Kawaguchi, 2016), and certain positively homogeneous networks (Haeffele & Vidal, 2015; 2017) have such a landscape property, but unfortunately condition 2) does not hold for networks with multiple hidden layers (Kawaguchi, 2016). Moreover, the landscape-based analysis generally fails to provide a good characterization of the convergence rate, except for a local rate around the equilibrium (Lee et al., 2016; Ge et al., 2017). In fact, during early stages of training, gradient descent could take exponential time to escape some saddle points if not initialized properly (Du et al., 2017).

The *trajectory-based* analyses study the training dynamics of the weights given a specific initialization. For example, the case of small initialization has been studied for various models (Arora et al., 2019a; Gidel et al., 2019; Li et al., 2018b; Stöger & Soltanolkotabi, 2021; Li et al., 2021b;a). Under this type of initialization, the trained model is implicitly biased towards low-rank (Arora et al., 2019a; Gidel et al., 2019; Li et al., 2018b; Stöger & Soltanolkotabi, 2021; Li et al., 2021b), and sparse (Li et al., 2021a) models. While the analysis for small initialization gives rich insights on the generalization of neural networks, the number of iterations required for gradient descent to find a good model often increases as the initialization scale decreases. Such dependence proves to be logarithmic on the scale for symmetric matrix factorization model (Li et al., 2018b; Stöger & Soltanolkotabi, 2021; Li et al., 2021b), but for deep networks, existing analysis at best shows a polynomial dependency (Li et al., 2021a). Therefore, the analysis for small initialization, while insightful in understanding the implicit bias of neural network training, is not suitable for understanding the training efficiency in practice since small initialization is rarely implemented due to its slow convergence. Another line of work studies the initialization in the kernel regime, where a randomly initialized sufficiently wide neural

network can be well approximated by its linearization at initialization Jacot et al. (2018); Chizat et al. (2019); Arora et al. (2019b). In this regime, gradient descent enjoys a linear rate of convergence toward the global minimum (Du et al., 2019; Allen-Zhu et al., 2019; Du & Hu, 2019). However, the width requirement in the analysis is often unrealistic, and empirical evidence has shown that practical neural networks generally do not operate in the kernel regime (Chizat et al., 2019).

The study of non-small, non-kernel-regime initialization has been mostly centered around linear models. For matrix factorization models, spectral initialization (Saxe et al., 2014; Gidel et al., 2019; Tarmoun et al., 2021) allows for decoupling the training dynamics into several scalar dynamics. For non-spectral initialization, the notion of weight *imbalance*, a quantity that depends on the differences between the weights matrices of adjacent layers, is crucial in most analyses. When the initialization is balanced, i.e., when the imbalance matrices are zero, the convergence relies on the initial end-to-end linear model being close to its optimum (Arora et al., 2018a;b). It has been shown that having a non-zero imbalance potentially improves the convergence rate (Tarmoun et al., 2021; Min et al., 2021), but the analysis only works for two-layer models. For deep linear networks, the effect of weight imbalance on the convergence has been only studied in the case when all imbalance matrices are positive semi-definite (Yun et al., 2020), which is often unrealistic in practice. Lastly, most of the aforementioned analyses study the $l_2$ loss for regression tasks, and it remains unknown whether they can be generalized to other types of losses commonly used in classification tasks.

**Our contribution**: This paper aims to provide a general framework for analyzing the convergence of gradient flow on multi-layer linear models. We consider the gradient flow on a loss function of the form $\mathcal{L} = f(W_1 W_2 \cdots W_L)$, where $f$ satisfies the gradient dominance property. We show that with proper initialization, the loss converges to its global minimum exponentially. More specifically:

- Our analysis shows that the convergence rate depends on two trajectory-specific quantities: 1) the *imbalance matrices*, which measure the difference between the weights of adjacent layers, and 2) a lower bound on the least singular values of *weight product* $W = W_1 W_2 \cdots W_L$. The former is time-invariant under gradient flow, thus it is fully determined by the initialization, while the latter can be controlled by initializing the product sufficiently close to its optimum.

- Our analysis covers most initialization schemes used in prior work (Saxe et al., 2014; Tarmoun et al., 2021; Arora et al., 2018a;b; Min et al., 2021; Yun et al., 2020) for both multi-layer linear networks and diagonal linear networks while providing convergence guarantees for a wider range of initializations. Furthermore, our rate bounds characterize the general effect of weight imbalance on convergence.

- Our convergence results directly apply to loss functions commonly used in regression tasks, and can be extended to loss functions used in classification tasks with an alternative assumption on $f$, under which we show $\mathcal{O}(1/t)$ convergence of the loss.

**Notations**: For an $n \times m$ matrix $A$, we let $A^T$ denote the matrix transpose of $A$, $\sigma_i(A)$ denote its $i$-th singular value in decreasing order and we conveniently write $\sigma_{\min}(A) = \sigma_{\min\{n,m\}}(A)$ and let $\sigma_k(A) = 0$ if $k > \min\{n, m\}$. We also let $\|A\|_2 = \sigma_1(A)$ and $\|A\|_F = \sqrt{\operatorname{tr}(A^T A)}$. For a square matrix of size $n$, we let $\operatorname{tr}(A)$ denote its trace and we let $\operatorname{diag}\{a_i\}_{i=1}^n$ be a diagonal matrix with $a_i$ specifying its $i$-th diagonal entry. For a Hermitian matrix $A$ of size $n$, we let $\lambda_i(A)$ denote its $i$-th eigenvalue and we write $A \succeq 0$ ($A \preceq 0$) when $A$ is positive semi-definite (negative semi-definite). For two square matrices $A, B$ of the same size, we let $\langle A, B \rangle_F = \operatorname{tr}(A^T B)$. For a scalar-valued or matrix-valued function of time, $F(t)$, we write $\dot{F}, \dot{F}(t)$ or $\frac{d}{dt} F(t)$ for its time derivative. Additionally, we use $I_n$ to denote the identity matrix of order $n$ and $\mathcal{O}(n)$ to denote the set of $n \times n$ orthogonal matrices. Lastly, we use $[\cdot]_+ := \max\{\cdot, 0\}$.

## 2 OVERVIEW OF THE ANALYSIS

This paper considers the problem of finding a matrix $W$ that solves

$$\min_{W \in \mathbb{R}^{n \times m}} f(W), \tag{1}$$

with the following assumption on $f$.

**Assumption 1.** *The function $f$ is differentiable and satisfies[1]:*

**A1:** *$f$ satisfies the Polyak-Łojasiewicz (PL) condition, i.e. $\|\nabla f(W)\|_F^2 \geq \gamma(f(W) - f^*), \forall W$. This condition is also known as gradient dominance.*

**A2:** *$f$ is $K$-smooth, i.e., $\|\nabla f(W) - \nabla f(V)\|_F \leq K\|W - V\|_F, \forall W, V$, and $f$ is $\mu$-strongly convex, i.e., $f(W) \geq f(V) + \langle \nabla f(V), W - V \rangle_F + \frac{\mu}{2}\|W - V\|_F^2, \forall W, V$.*

While classic work (Polyak, 1987) has shown that the gradient descent update on $W$ with proper step size ensures a linear rate of convergence of $f(W)$ towards its optimal value $f^*$, the recent surge of research on the convergence and implicit bias of gradient-based methods for deep neural networks has led to a great amount of work on the *overparametrized* problem:

$$\min_{\{W_l\}_{l=1}^L} \mathcal{L}\left(\{W_l\}_{l=1}^L\right) = f(W_1 W_2 \cdots W_L), \tag{2}$$

where $L \geq 2$, $W_l \in \mathbb{R}^{h_{l-1} \times h_l}, i = 1, \cdots, L$, with $h_0 = n, h_L = m$ and $\min\{h_1, \cdots, h_{L-1}\} \geq \min\{n, m\}$. This assumption on $\min\{h_1, \cdots, h_{L-1}\}$ is necessary to ensure that the optimal value of (2) is also $f^*$, and in this case, the *product* $\prod_{l=1}^L W_l$ can represent an *overparametrized linear network/model* (Arora et al., 2018b; Tarmoun et al., 2021; Min et al., 2021)

## 2.1 CONVERGENCE VIA GRADIENT DOMINANCE

For problem (2), consider the gradient flow dynamics on the loss function $\mathcal{L}\left(\{W_l\}_{l=1}^L\right)$:

$$\dot{W}_l = -\frac{\partial}{\partial W_l}\mathcal{L}\left(\{W_l\}_{l=1}^L\right), l = 1, \cdots, L. \tag{3}$$

The gradient flow dynamics can be viewed as gradient descent with "infinitesimal" step size and convergence results for gradient flow can be used to understand the corresponding gradient descent algorithm with sufficiently small step size (Elkabetz & Cohen, 2021). We have the following result regarding the time-derivative of $\mathcal{L}$ under gradient flow (3).

**Lemma 1.** *Under continuous dynamics in* (3)*, we have*

$$\dot{\mathcal{L}} = -\|\nabla\mathcal{L}\left(\{W_l\}_{l=1}^L\right)\|_F^2 = -\left\langle \mathcal{T}_{\{W_l\}_{l=1}^L}\nabla f(W), \nabla f(W)\right\rangle_F, \tag{4}$$

*where $W = \prod_{l=1}^L W_l$, and $\mathcal{T}_{\{W_l\}_{l=1}^L}$ is the following positive semi-definite linear operator on $\mathbb{R}^{n \times m}$*

$$\mathcal{T}_{\{W_l\}_{l=1}^L}E = \sum_{l=1}^L \left(\prod_{i=0}^{l-1} W_i\right)\left(\prod_{i=0}^{l-1} W_i\right)^T E \left(\prod_{i=l+1}^{L+1} W_i\right)^T \left(\prod_{i=l+1}^{L+1} W_i\right), W_0 = I_n, W_{L+1} = I_m.$$

Such an expression of $\|\nabla\mathcal{L}\|_F^2$ has been studied in Arora et al. (2018b), and we include a proof in Appendix C for completeness. Our convergence analysis is as follows.

For this overparameterized problem, the minimum $\mathcal{L}^*$ of (2) is $f^*$. Then from Lemma 1 and Assumption **A1**, we have

$$\begin{aligned}
\dot{\mathcal{L}} = &-\left\langle \mathcal{T}_{\{W_l\}_{l=1}^L}\nabla f(W), \nabla f(W)\right\rangle_F \\
&\leq -\lambda_{\min}(\mathcal{T}_{\{W_l\}_{l=1}^L})\|\nabla f(W)\|_F^2 \qquad \text{(min-max theorem (Teschl, 2014))} \tag{5} \\
&\overset{(A1)}{\leq} -\lambda_{\min}(\mathcal{T}_{\{W_l\}_{l=1}^L})\gamma(f(W) - f^*) = -\lambda_{\min}(\mathcal{T}_{\{W_l\}_{l=1}^L})\gamma(\mathcal{L} - \mathcal{L}^*).
\end{aligned}$$

If we can find a lower bound $\alpha > 0$ such that $\lambda_{\min}(\mathcal{T}_{\{W_l(t)\}_{l=1}^L}) \geq \alpha, \forall t \geq 0$, then the following inequality holds on the entire training trajectory $\frac{d}{dt}(\mathcal{L} - \mathcal{L}^*) \leq -\alpha\gamma(\mathcal{L} - \mathcal{L}^*)$. Therefore, by using Grönwall's inequality (Grönwall, 1919), we can show that the loss function $\mathcal{L}$ converges exponential to its minimum, i.e.,

$$\mathcal{L}(t) - \mathcal{L}^* \leq \exp(-\alpha\gamma t)(\mathcal{L}(0) - \mathcal{L}^*), \forall t \geq 0. \tag{6}$$

---

[1]Note that **A2** assumes $\mu$-strong convexity, which implies **A1** with $\gamma = 2\mu$. However, we list **A1** and **A2** separately since they have different roles in our analysis.

Therefore, to show exponential convergence of the loss, we need to lower bound $\lambda_{\min}(\mathcal{T}_{\{W_l(t)\}_{l=1}^L})$. Most existing work on the convergence of gradient flow/descent on linear networks implicitly provides such a lower bound, given additional assumptions on the initialization $\{W_l(0)\}_{l=1}^L$, though not presented with such generality. We revisit previous analyses to see how such a problem can be solved for two-layer linear networks, then present our new results regarding deep linear networks.

## 3 Lessons from Two-layer Linear Models

In this section, we revisit prior work through the lens of our general convergence analysis in Section 2.1. A lower bound on $\lambda_{\min}(\mathcal{T}_{\{W_l(t)\}_{l=1}^L})$ can be obtained from the training invariance of the gradient flow. We first consider the following *imbalance matrices*:

$$D_l := W_l^T W_l - W_{l+1} W_{l+1}^T, \ l = 1, \cdots, L-1. \tag{7}$$

For such imbalance matrices, we have

**Lemma 2.** *Under the continuous dynamics* (3), *we have* $\dot{D}_l(t) = 0, \forall t \geq 0, l = 1, \cdots, L-1$.

Such invariance of weight imbalance has been studied in most work on linear networks (Arora et al., 2018a; Du et al., 2018; Yun et al., 2020). We include the proof in Appendix C for completeness. Since the imbalance matrices $\{D_l\}_{l=1}^{L-1}$ are fixed at its initial value, any point $\{W_l(t)\}_{l=1}^L$ on the training trajectory must satisfy the *imbalance constraints* $W_l(t)^T W_l(t) - W_{l+1} W_{l+1}^T = D_l(0), \ l = 1, \cdots, L-1$. Previous work has shown that enforcing certain non-zero imbalance at initialization leads to exponential convergence of the loss for two-layer networks (Tarmoun et al., 2021; Min et al., 2021), and for deep networks (Yun et al., 2020). Another line of work (Arora et al., 2018a;b) has shown that balanced initialization ($D_l = 0, \forall l$) have exactly $\lambda_{\min}(\mathcal{T}_{\{W_l(t)\}_{l=1}^L}) = L\sigma_{\min}^{2-2/L}(W(t))$, where $W(t) = \prod_{l=1}^L W_l(t)$. This suggests that the bound on $\lambda_{\min}(\mathcal{T}_{\{W_l(t)\}_{l=1}^L})$ we are looking for should potentially depend on both the *weight imbalance* matrices and *weight product* matrix.

Indeed, for two-layer models, a re-statement[2] of the results in (Min et al., 2022) provides a lower bound on $\lambda_{\min}(\mathcal{T}_{\{W_1, W_2\}})$ with the knowledge of the imbalance and the product.

**Lemma 3** (re-stated from Min et al. (2022)). *When* $L = 2$, *given weights* $\{W_1, W_2\}$ *with imbalance matrix* $D = W_1^T W_1 - W_2 W_2^T$ *and product* $W = W_1 W_2$, *define*

$$\Delta_+ = [\lambda_1(D)]_+ - [\lambda_n(D)]_+ \,, \Delta_- = [\lambda_1(-D)]_+ - [\lambda_m(-D)]_+ \,, \underline{\Delta} = [\lambda_n(D)]_+ + [\lambda_m(-D)]_+ \,. \tag{8}$$

*Then for the linear operator* $\mathcal{T}_{\{W_1, W_2\}}$ *defined in Lemma 1, we have*

$$\lambda_{\min}(\mathcal{T}_{\{W_1, W_2\}}) \geq \frac{1}{2}\Big( -\Delta_+ + \sqrt{(\Delta_+ + \underline{\Delta})^2 + 4\sigma_n^2(W)} - \Delta_- + \sqrt{(\Delta_- + \underline{\Delta})^2 + 4\sigma_m^2(W)} \Big). \tag{9}$$

Min et al. (2022) include a detailed discussion on the bound, including tightness. For our purpose, we note the following:

**Effect of imbalance**: It follows from (9) that $\lambda_{\min}(\mathcal{T}_{\{W_1, W_2\}}) \geq \underline{\Delta}$ since $\sigma_{\min}(W) \geq 0$. Therefore, $\underline{\Delta}$ is always a lower bound on the convergence rate. This means that, for most initializations, the fact that the imbalance matrices are bounded away from zero (characterized by $\underline{\Delta} > 0$) is already sufficient for exponential convergence.

**Effect of product**: The role of the product in (9) is more nuanced: Assume $n = m$ for simplicity so that $\sigma_n(WW^T) = \sigma_m(W^TW) = \sigma_{\min}^2(W)$. We see that the non-negative quantities $\Delta_+, \Delta_-$ control how much the product affects the convergence. More precisely, the lower bound in (9) is a decreasing function of both $\Delta_+$ and $\Delta_-$. When $\Delta_+ = \Delta_- = 0$, the lower bound reduces to $\sqrt{\underline{\Delta}^2 + 4\sigma_{\min}^2(W)}$, showing a joint contribution to convergence from both imbalance and product. However, as $\Delta_+, \Delta_-$ increases, the bound decreases towards $\underline{\Delta}$, which means that the effect of

---

[2] In Min et al. (2022), there is no general idea of lower bounding $\lambda_{\min}(\mathcal{T}_{\{W_1, W_2\}})$, but their analyses essentially provide such a bound.

imbalance always exists, but the effect of the product diminishes for large $\Delta_+, \Delta_-$. We note that $\Delta_+, \Delta_-$ measure how the eigenvalues of the imbalance matrix $D$ are different in magnitude, i.e., how "ill-conditioned" the imbalance matrix is.

**Implication on convergence**: Note that (9) is almost a lower bound for $\lambda_{\min}\left(\mathcal{T}_{\{W_1(t), W_2(t)\}}\right), t \geq 0$, as the imbalance matrix $D$ is time-invariant (so are $\Delta_+, \Delta_-, \underline{\Delta}$), except the right-hand side of (9) also depends on $\sigma_{\min}(W(t))$. If $f$ satisfies **A2**, then $f$ has a unique minimizer $W^*$. Moreover, one can show that given a initial product $W(0)$, $W(t)$ is constrained to lie within a closed ball $\left\{W : \|W - W^*\|_F \leq \sqrt{\frac{K}{\mu}}\|W(0) - W^*\|_F\right\}$. That is, the product $W(t)$ does not get too far away from $W^*$ during training. We can use this to derive the following lower bound on $\sigma_{\min}(W(t))$:

$$\sigma_{\min}(W(t)) \geq \left[\sigma_{\min}(W^*) - \sqrt{\frac{K}{\mu}}\|W(0) - W^*\|_F\right]_+ := margin \quad \text{(See Appendix A).} \quad (10)$$

This margin term being positive guarantees that the closed ball excludes any $W$ with $\sigma_{\min}(W) = 0$. With this observation, we find a lower bound $\lambda_{\min}\left(\mathcal{T}_{\{W_1(t), W_2(t)\}}\right), t \geq 0$ that depends on both the weight imbalance and margin, and the exponential convergence of loss $\mathcal{L}$ follows:

**Theorem 1.** *Let $D$ be the imbalance matrix for $L = 2$. The continuous dynamics in* (3) *satisfy*

$$\mathcal{L}(t) - \mathcal{L}^* \leq \exp\left(-\alpha_2 \gamma t\right)\left(\mathcal{L}(0) - \mathcal{L}^*\right), \forall t \geq 0, \quad (11)$$

*where*

1. *If $f$ satisfies only **A1**, then $\alpha_2 = \underline{\Delta}$;*

2. *If $f$ satisfies both **A1** and **A2**, then*

$$\alpha_2 = -\Delta_+ + \sqrt{(\Delta_+ + \underline{\Delta})^2 + 4\left(\left[\sigma_n\left(W^*\right) - \sqrt{K/\mu}\|W(0) - W^*\|_F\right]_+\right)^2}$$

$$- \Delta_- + \sqrt{(\Delta_- + \underline{\Delta})^2 + 4\left(\left[\sigma_m\left(W^*\right) - \sqrt{K/\mu}\|W(0) - W^*\|_F\right]_+\right)^2}, \quad (12)$$

*with $W(0) = \prod_{l=1}^{L} W_l(0)$ and $W^*$ equal to the unique optimizer of $f$.*

Please see Appendix E for the proof. Theorem 1 is new as it generalizes the convergence result in Min et al. (2022) for two-layer linear networks, which is only for $l_2$ loss in linear regression. Our result considers a general loss function defined by $f$, including the losses for matrix factorization (Arora et al., 2018a), linear regression (Min et al., 2022), and matrix sensing (Arora et al., 2019a). Additionally, Arora et al. (2018a) first introduced the notion of margin for $f$ in matrix factorization problems ($K = 1, \mu = 1$), and we extend it to any $f$ that is smooth and strongly convex.

**Towards deep models**: So far, we revisited prior results on two-layer networks, showing how $\lambda_{\min}(\mathcal{T}_{W_1, W_2})$ can be lower bounded by weight imbalance and product, from which the convergence result is derived. Can we generalize the analysis to deep networks? The main challenge is that even computing $\lambda_{\min}(\mathcal{T}_{\{W_l\}_{l=1}^L})$ given the weights $\{W_l\}_{l=1}^L$ is complicated: For $L = 2$, $\lambda_{\min}(\mathcal{T}_{W_1, W_2}) = \lambda_n(W_1 W_1^T) + \lambda_m(W_2^T W_2)$, but such nice relation does not exist for $L > 3$, which makes the search for a tight lower bound as in (9) potentially difficult. On the other hand, the findings in (9) shed light on what can be potentially shown for the deep layer case:

1. For two-layer networks, we always have the bound $\lambda_{\min}\left(\mathcal{T}_{\{W_1, W_2\}}\right) \geq \underline{\Delta}$, which depends only on the imbalance. *Can we find a lower bound on the convergence rate of a deep network that depends only on an imbalance quantity analogous to $\underline{\Delta}$? If yes, how does such a quantity depend on network depth?*

2. For two-layer networks, the bound reduces to $\sqrt{\underline{\Delta}^2 + 4\sigma_{\min}^2(W)}$ when the imbalance is "well-conditioned" ($\Delta_+, \Delta_-$ are small). *For deep networks, can we characterize such joint contribution from the imbalance and product, given a similar assumption?*

We will answer these questions as we present our convergence results for deep networks.

## 4 CONVERGENCE RESULTS FOR DEEP LINEAR MODELS

### 4.1 THREE-LAYER MODEL

Beyond two-layer models, the convergence analysis for imbalanced networks not in the kernel regime has only been studied for specific initializations (Yun et al., 2020). In this section, we derive a novel rate bound for three-layer models that applies to a wide range of imbalanced initializations. For ease of presentation, we denote the two imbalance matrices for three-layer models, $D_1$ and $D_2$, as

$$-D_1 = W_2 W_2^T - W_1^T W_1 := D_{21}, \quad D_2 = W_2^T W_2 - W_3 W_3^T := D_{23}. \tag{13}$$

Our lower bound on $\lambda_{\min}\left(\mathcal{T}_{\{W_1, W_2, W_3\}}\right)$ comes after a few definitions.

**Definition 1.** *Given two real symmetric matrices $A, B$ of order $n$, we define the non-commutative binary operation $\wedge_r$ as $A \wedge_r B := \text{diag}\{\min\{\lambda_i(A), \lambda_{i+1-r}(B)\}\}_{i=1}^n$, where $\lambda_j(\cdot) = +\infty, \forall j \leq 0$.*

**Definition 2.** *Given imbalance matrices $(D_{21}, D_{23}) \in \mathbb{R}^{h_1 \times h_1} \times \mathbb{R}^{h_2 \times h_2}$, define*

$$\bar{D}_{h_1} = \text{diag}\{\max\{\lambda_i(D_{21}), \lambda_i(D_{23}), 0\}\}_{i=1}^{h_1}, \quad \bar{D}_{h_2} = \text{diag}\{\max\{\lambda_i(D_{21}), \lambda_i(D_{23}), 0\}\}_{i=1}^{h_2}, \tag{14}$$

$$\Delta_{21} = \text{tr}(\bar{D}_{h_1}) - \text{tr}(\bar{D}_{h_1} \wedge_n D_{21}), \qquad \Delta_{21}^{(2)} = \text{tr}(\bar{D}_{h_1}^2) - \text{tr}\left((\bar{D}_{h_1} \wedge_n D_{21})^2\right), \tag{15}$$

$$\Delta_{23} = \text{tr}(\bar{D}_{h_2}) - \text{tr}(\bar{D}_{h_2} \wedge_m D_{23}), \qquad \Delta_{23}^{(2)} = \text{tr}(\bar{D}_{h_2}^2) - \text{tr}\left((\bar{D}_{h_2} \wedge_m D_{23})^2\right). \tag{16}$$

**Theorem 2.** *When $L = 3$, given weights $\{W_1, W_2, W_3\}$ with imbalance matrices $(D_{21}, D_{23})$, then for the linear operator $\mathcal{T}_{\{W_1, W_2, W_3\}}$ defined in Lemma 1, we have*

$$\lambda_{\min}\left(\mathcal{T}_{\{W_1, W_2, W_3\}}\right) \geq \frac{1}{2}(\Delta_{21}^{(2)} + \Delta_{21}^2) + \Delta_{21}\Delta_{23} + \frac{1}{2}(\Delta_{23}^{(2)} + \Delta_{23}^2) \tag{17}$$

*Proof Sketch.* Generally, it is difficult to directly work on $\lambda_{\min}\left(\mathcal{T}_{\{W_1, W_2, W_3\}}\right)$, and we use the lower bound $\lambda_{\min}\left(\mathcal{T}_{\{W_1, W_2, W_3\}}\right) \geq \lambda_n(W_1 W_2 W_2^T W_1^T) + \lambda_n(W_1 W_1^T)\lambda_m(W_3^T W_3) + \lambda_m(W_3^T W_2^T W_2 W_3)$. We show that given $D_{21}, D_{23}$, the optimal value of

$$\min_{W_1, W_2, W_3} \lambda_n(W_1 W_2 W_2^T W_1^T) + \lambda_n(W_1 W_1^T)\lambda_m(W_3^T W_3) + \lambda_m(W_3^T W_2^T W_2 W_3) \tag{18}$$

$$s.t. \quad W_2 W_2^T - W_1^T W_1 = D_{21}, \qquad W_2^T W_2 - W_3 W_3^T = D_{23}$$

is $\Delta^*(D_{21}, D_{23}) = \frac{1}{2}(\Delta_{21}^{(2)} + \Delta_{21}^2) + \Delta_{21}\Delta_{23} + \frac{1}{2}(\Delta_{23}^{(2)} + \Delta_{23}^2)$, the bound shown in (17). Please see Appendix F for the complete proof and a detailed discussion on the proof idea.

□

With the theorem we immediately have the following corollary.

**Corollary 1.** *When $L = 3$, given initialization with imbalance matrices $(D_{21}, D_{23})$ and $f$ satisfying A1, the continuous dynamics in (3) satisfy*

$$\mathcal{L}(t) - \mathcal{L}^* \leq \exp\left(-\alpha_3 \gamma t\right)\left(\mathcal{L}(0) - \mathcal{L}^*\right), \forall t \geq 0, \tag{19}$$

*where $\alpha_3 = \frac{1}{2}(\Delta_{21}^{(2)} + \Delta_{21}^2) + \Delta_{21}\Delta_{23} + \frac{1}{2}(\Delta_{23}^{(2)} + \Delta_{23}^2)$.*

We make the following remarks regarding the contribution.

**Optimal bound via imbalance**: First of all, as shown in the proof sketch, our bound should be considered as the best lower bound on $\lambda_{\min}(\mathcal{T}_{\{W_1(t), W_2(t), W_3(t)\}})$ one can obtain given knowledge of the imbalance matrices $D_{21}$ and $D_{23}$ only. More importantly, this lower bound works for ANY initialization and has the same role as $\underline{\Delta}$ does in two-layer linear networks, i.e., (17) quantifies the general effect imbalance on the convergence. Finding an improved bound that takes the effect of product $\sigma_{\min}(W)$ into account is an interesting future research direction.

**Implication on convergence**: Corollary 2 shows exponential convergence of the loss $\mathcal{L}(t)$ if $\alpha_3 > 0$. While it is challenging to characterize all initialization such that $\alpha_3 > 0$, the case $n = m = 1$ is rather simpler: In this case, $\bar{D}_{h_1} \wedge_1 D_{21} = D_{21}$ and $\bar{D}_{h_2} \wedge_1 D_{23} = D_{23}$. Then we have

$$\Delta_{21} = \text{tr}(\bar{D}_{h_1}) - \text{tr}(D_{21}) = \sum_{i=1}^{h_1}(\lambda_i(\bar{D}_{h_1}) - \lambda_i(D_{21})) + \lambda_{h_1}(\bar{D}_{h_1}) - \lambda_{h_1}(D_{21}) \geq -\lambda_{h_1}(D_{21}),$$

and similarly we have $\Delta_{23} \geq -\lambda_{h_2}(D_{23})$. Therefore, $\alpha_3 \geq \Delta_{21}\Delta_{23} \geq \lambda_{h_1}(D_{21})\lambda_{h_2}(D_{23}) > 0$ when both $D_{21}$ and $D_{23}$ have negative eigenvalues, which is easy to satisfy as both $D_{21}$ and $D_{23}$ are given by the difference between two positive semi-definite matrices. Such observation can be generalized to show that $\alpha_3 > 0$ when $D_{21}$ has at least $n$ negative eigenvalues and $D_{23}$ has at least $m$ negative eigenvalues. Moreover, we show that $\alpha_3 > 0$ under certain definiteness assumptions on $D_{21}$ and $D_{23}$, please refer to the remark after Theorem 3 in Section 4.2. A better characterization of the initialization that has $\alpha_3 > 0$ is an interesting future research topic.

**Technical contribution**: The way we find the lower bound in (17) is by studying the generalized eigenvalue interlacing relation imposed by the imbalance constraints. Specifically, $W_2 W_2^T - W_1^T W_1 = D_{21}$ suggests that $\lambda_{i+n}(W_2 W_2^T) \leq \lambda_i(D_{21}) \leq \lambda_i(W_2 W_2^T), \forall i$ because $W_2 W_2^T - D_{21}$ is a matrix of at most rank-$n$. We derive, from such interlacing relation, novel eigenvalue bounds (See Lemma F.6) on $\lambda_n(W_1^T W_1)$ and $\lambda_n(W_1 W_2 W_2^T W_1)$ that depends on eigenvalues of both $W_2 W_2^T$ and $D_{21}$. Then the eigenvalues of $W_2 W_2^T$ can also be controlled by the fact that $W_2$ must satisfy both imbalance equations in (13). Since imbalance equations like those in (13) appear in deep networks and certain nonlinear networks Du et al. (2018); Le & Jegelka (2022), we believe our mathematical results are potentially useful for understanding those networks.

**Comparison with prior work**: The convergence of multi-layer linear networks under balanced initialization ($D_l = 0, \forall l$) has been studied in Arora et al. (2018a;b), and our result is complementary as we study the effect of non-zero imbalance on the convergence of three-layer networks. Some settings with imbalanced weights have been studied: Yun et al. (2020) studies a special initialization scheme ($D_l \succeq 0, l = 1, \cdots, L-2$, and $D_{L-1} \succeq \lambda I_{h_{L-1}}$) that forces the partial ordering of the weights, and Wu et al. (2019) uses a similar initialization to study the linear residual networks. Our bound works for such initialization and also show such partial ordering is not necessary for convergence.

## 4.2 Deep Linear Models

The lower bound we derived for three-layer networks applies to any initialization. However, the bound is a fairly complicated function of all the imbalance matrices that is hard to interpret. Searching for such a general bound is even more challenging for models with arbitrary depth ($L \geq 3$). Therefore, our results for deep networks will rely on extra assumptions on the weights that simplify the lower bound to facilite interpretability. Specifically, we consider the following properties of the weights:

**Definition 3.** *A set of weights* $\{W_l\}_{l=1}^L$ *with imbalance matrices* $\{D_l := W_l^T W_l - W_{l+1} W_{l+1}^T\}_{l=1}^{L-1}$ *is said to be **unimodal with index** $l^*$ if there exists some $l^* \in [L]$ such that*

$$D_l \succeq 0, \quad for \ l < l^* \qquad and \qquad D_l \preceq 0, \quad for \ l \geq l^*.$$

*We define its **cumulative imbalances** $\{\tilde{d}_{(i)}\}_{i=1}^{L-1}$ as* $\tilde{d}_{(i)} = \begin{cases} \sum_{l=l^*}^{i} \lambda_m(-D_l), & i \geq l^* \\ \sum_{l=i}^{l^*-1} \lambda_n(D_l), & i < l^* \end{cases}.$

*Furthermore, for weights with unimodality index $l^*$, if additionally, $D_l = d_l I_{h_l}, l = 1, \cdots, L-1$ for*

$$d_l \geq 0, \quad for \ l < l^* \qquad and \qquad d_l \leq 0, \quad for \ l \geq l^*,$$

*then those weights are said to have **homogeneous imbalance**.*

The unimodality assumption enforces an ordering of the weights w.r.t. the positive semi-definite cone. This is more clear when considering scalar weights $\{w_l\}_{l=1}^L$, in which case unimodality requires $w_l^2$ to be descending until index $l^*$ and ascending afterward. Under this unimodality assumption, we show that imbalance contributes to the convergence of the loss via a product of cumulative imbalanaces. Furthermore, we also show the combined effects of imbalance and weight product when the imbalance matrices are "well-conditioned" (in this case, homogeneous). More formally, we have:

**Theorem 3.** *For weights* $\{W_l\}_{l=1}^L$ *with unimodality index $l^*$, we have*

$$\lambda_{\min}\left(\mathcal{T}_{\{W_l\}_{l=1}^L}\right) \geq \prod_{l=1}^{L-1} \tilde{d}_{(i)}. \tag{20}$$

*Furthermore, if the weights have homogeneous imbalance, then*

$$\lambda_{\min}\left(\mathcal{T}_{\{W_l\}_{l=1}^L}\right) \geq \sqrt{\left(\prod_{l=1}^{L-1} \tilde{d}_{(i)}\right)^2 + \left(L\sigma_{\min}^{2-2/L}(W)\right)^2}, \quad W = \prod_{l=1}^L W_l. \tag{21}$$

We make the following remarks:

**Connection to results for three-layer**: For three-layer networks, we present an optimal bound

$$\lambda_{\min}(\mathcal{T}_{W_1,W_2,W_3}) \geq \frac{1}{2}(\Delta_{21}^{(2)} + \Delta_{21}^2) + \Delta_{21}\Delta_{23} + \frac{1}{2}(\Delta_{23}^{(2)} + \Delta_{23}^2),$$

given knowledge of the imbalance. Interestingly, when comparing it with our bound in (20), we have:

**Claim.** *When $L = 3$, for weights $\{W_1, W_2, W_3\}$ with unimodality index $l^*$,*

1. *If $l^* = 1$, then $\frac{1}{2}(\Delta_{23}^{(2)} + \Delta_{23}^2) = \prod_{l=1}^{L-1} \tilde{d}_{(i)}$ and $\frac{1}{2}(\Delta_{21}^{(2)} + \Delta_{21}^2) = \Delta_{21}\Delta_{23} = 0$;*

2. *If $l^* = 2$, then $\Delta_{21}\Delta_{23} = \prod_{l=1}^{L-1} \tilde{d}_{(i)}$ and $\frac{1}{2}(\Delta_{21}^{(2)} + \Delta_{21}^2) = \frac{1}{2}(\Delta_{23}^{(2)} + \Delta_{23}^2) = 0$;*

3. *If $l^* = 3$, then $\frac{1}{2}(\Delta_{21}^{(2)} + \Delta_{21}^2) = \prod_{l=1}^{L-1} \tilde{d}_{(i)}$ and $\frac{1}{2}(\Delta_{23}^{(2)} + \Delta_{23}^2) = \Delta_{21}\Delta_{23} = 0$.*

We refer the readers to Appendix G for the proof. The claim shows that the bound in (20) is optimal for three-layer unimodal weights as it coincides with the one in Theorem 2. We conjecture that (20) is also optimal for multi-layer unimodal weights and leave the proof for future research. Interestingly, while the bound for three-layer models is complicated, the three terms $\frac{1}{2}(\Delta_{23}^{(2)} + \Delta_{23}^2)$, $\Delta_{21}\Delta_{23}$, $\frac{1}{2}(\Delta_{21}^{(2)} + \Delta_{21}^2)$, seem to roughly capture how close the weights are to those with unimodality. This hints at potential generalization of Theorem 2 to the deep case where the bound should have $L$ terms capturing how close the weights are to those with different unimodality ($l^* = 1, \cdots, L$).

**Effect of imbalance under unimodality**: For simplicity, we assume unimodality index $l^* = L$. The bound $\prod_{i=1}^{L-1} \tilde{d}_{(i)}$, as a product of cumulative imbalances, generally grows exponentially with the depth $L$. Prior work Yun et al. (2020) studies the case $D_l \succeq 0, l = 1, \cdots, L-2$, and $D_{L-1} \succeq \lambda I_{h_{L-1}}$, in which case $\prod_{i=1}^{L-1} \tilde{d}_{(i)} \geq \lambda^{L-1}$. Our bound $\prod_{i=1}^{L-1} \tilde{d}_{(i)}$ suggests the dependence on $L$ could be super-exponential: When $\lambda_n(D_l) \geq \epsilon > 0$, for $l = 1, \cdots, L-1$, we have $\prod_{i=1}^{L-1} \tilde{d}_{(i)} = \prod_{i=1}^{L-1} \sum_{l=i}^{L-1} \lambda_n(D_l) \geq \prod_{l=1}^{L-1} l\epsilon = \epsilon^{L-1}(L-1)!$, which grows faster in $L$ than $\lambda^{L-1}$ for any $\lambda$. Therefore, for gradient flow dynamics, the depth $L$ could greatly improve convergence in the presence of weight imbalance. One should note, however, that such analysis can not be directly translated into fast convergence guarantees of gradient descent algorithm as one requires careful tuning of the step size for the discrete weight updates to follow the trajectory of the continuous dynamics (Elkabetz & Cohen, 2021).

With our bound in Theorem 3, we show convergence of deep linear models under various initialization:

**Convergence under unimodality**: The following immediately comes from Theorem 3:

**Corollary 2.** *If the initialization weights $\{W_l(0)\}_{l=1}^L$ are unimodal, then the continuous dynamics in (3) satisfy*

$$\mathcal{L}(t) - \mathcal{L}^* \leq \exp\left(-\alpha_L \gamma t\right)(\mathcal{L}(0) - \mathcal{L}^*), \forall t \geq 0, \tag{22}$$

*where*

1. *If $f$ satisfies **A1** only, then $\alpha_L = \prod_{i=1}^{L-1} \tilde{d}_{(i)}$ ;*

2. *If $f$ satisfies both **A1**, **A2**, and the weights additionally have homogeneous imbalance, then*

$$\alpha_L = \sqrt{\left(\prod_{i=1}^{L-1} \tilde{d}_{(i)}\right)^2 + \left(L\left(\left[\sigma_{\min}(W^*) - \sqrt{K/\mu}\|W(0) - W^*\|_F\right]_+\right)^{2-2/L}\right)^2},$$

*with $W(0) = \prod_{l=1}^L W_l(0)$ and $W^*$ equal to the unique optimizer of $f$.*

**Spectral initialization under $l_2$ loss**: Suppose $f = \frac{1}{2}\|Y - W\|_F^2$ and $W = \prod_{l=1}^L W_l$. We write the SVD of $Y \in \mathbb{R}^{n \times m}$ as $Y = P \begin{bmatrix} \Sigma_Y & 0 \\ 0 & 0 \end{bmatrix} \begin{bmatrix} Q \\ 0 \end{bmatrix} := P\tilde{\Sigma}_Y \tilde{Q}$, where $P \in \mathcal{O}(n), Q \in \mathcal{O}(m)$. Consider the spectral initialization $W_1(0) = R\Sigma_1 V_1^T$, $W_l(0) = V_{l-1}\Sigma_l V_l^T$, $l = 2, \cdots, L-1$, $W_L(0) = V_{L-1}\Sigma_L \tilde{Q}$, where $\Sigma_l, l = 1, \cdots, L$ are diagonal matrices of our choice and $V_l \in \mathbb{R}^{n \times h_l}$, $l = 1, \cdots, L-1$ with $V_l^T V_l = I_{h_l}$. It can be shown that (See Appendix D.1 for details)

$$W_1(t) = R\Sigma_1(t)V_1^T, \quad W_l(t) = V_{l-1}\Sigma_l(t)V_l^T, \quad l = 2, \cdots, L-1, \quad W_L(t) = V_{L-1}\Sigma_L(t)\tilde{Q}. \tag{23}$$

Moreover, only the first $m$ diagonal entries of $\Sigma_l$ are changing. Let $\sigma_{i,l}, \sigma_{i,y}$ denote the $i$-th diagonal entry of $\Sigma_l$, and $\tilde{\Sigma}_Y$ respectively, then the dynamics of $\{\sigma_{i,l}\}_{l=1}^L$ follow the gradient flow on $\mathcal{L}_i(\{\sigma_{i,l}\}_{l=1}^L) = \frac{1}{2}\left|\sigma_{i,y} - \prod_{l=1}^L \sigma_{i,l}\right|^2$ for $i = 1, \cdots, m$, which is exactly a multi-layer model with scalar weights: $f(w) = |\sigma_{i,y} - w|^2/2, w = \prod_{l=1}^L w_l$. Therefore, spectral initialization under $l_2$ loss can be decomposed into $m$ deep linear models with scalar weights, whose convergence is shown by Corollary 2. Note that networks with scalar weights are always unimodal, because the gradient flow dynamics remain the same under any reordering of the weights, and always have homogeneous imbalance, because the imbalances are scalars. The aforementioned analysis also applies to the linear regression loss $f = \frac{1}{2}\|Y - XW\|_F^2$, provided that $\{X, Y\}$ is co-diagonalizable (Gidel et al., 2019), we refer the readers to Appendix D.1 for details.

**Diagonal linear networks**: Consider $f$ a function on $\mathbb{R}^n$ satisfying **A1** and $\mathcal{L} = f(w_1 \odot \cdots \odot w_L)$, where $w_l \in \mathbb{R}^n$ and $\odot$ denote the Hadamard (entrywise) product. The gradient flow on $\mathcal{L}$ can not be decomposed into several scalar dynamics as in the previous example, but we can show that (See Appendix D.2 for details) $\dot{\mathcal{L}} = -\|\nabla\mathcal{L}\|_F^2 \le -(\min_{1 \le i \le n} \lambda_{\min}(\mathcal{T}_{\{w_{l,i}\}_{l=1}^L}))\gamma(\mathcal{L} - \mathcal{L}^*)$, where $w_{l,i}$ is the $i$-th entry of $w_l$. Then Theorem 3 gives lower bound on each $\lambda_{\min}(\mathcal{T}_{\{w_{l,i}\}_{l=1}^L})$. Again, here the scalar weights $\{w_{l,i}\}_l^L$ always have homogeneous imbalance.

| Assumptions | Arora et al. (2018a) | Yun et al. (2020) | Ours |
|---|---|---|---|
| Unimodal weights | N/A | $\lambda^{L-1}$ | $\prod_{l=1}^{L-1} \tilde{d}(i)$ |
| Homogeneous imbalance | N/A | $\lambda^{L-1}$ | $\sqrt{(\prod_{l=1}^{L-1} \tilde{d}(i))^2 + (L\sigma_{\min}^{2-2/L}(W))^2}$ |
| Balanced | $L\sigma_{\min}^{2-2/L}(W)$ | N/A | |

Table 1: Compare our rate bound with prior work on deep networks.

**Comparison with prior work**: Regarding unimodality, Yun et al. (2020) studies the initialization scheme $D_l \succeq 0, l = 1, \cdots, L - 2$ and $D_{L-1} \succeq \lambda I_{h_{L-1}}$, which is a special case ($l^* = L$) of ours. The homogeneous imbalance assumption was first introduced in Tarmoun et al. (2021) for two-layer networks, and we generalize it to the deep case. We compare, in Table 1, our bound to the existing work (Arora et al., 2018a; Yun et al., 2020) on convergence of deep linear networks outside the kernel regime. Note that Yun et al. (2020) only studies a special case of unimodal weights ($l^* = L$ with $\tilde{d}_{(i)} \ge \lambda > 0, \forall i$). For homogeneous imbalance, Yun et al. (2020) studied spectral initialization and diagonal linear networks, whose initialization necessarily has homogeneous imbalance, but the result does not generalize to the case of matrix weights. Our results for homogeneous imbalance works also for deep networks with matrix weights, and our rate also shown the effect of the product $L\sigma_{\min}^{2-2/L}(W)$, thus covers the balanced initialization (Arora et al., 2018a) as well.

**Remark 1.** *Note that the loss functions used in Gunasekar et al. (2018); Yun et al. (2020) are classification losses, such as the exponential loss, which do not satisfy **A1**. However, they do satisfy Polyak-Łojasiewicz-inequality-like condition $\|\nabla f(W)\|_F \ge \gamma(f(W) - f^*), \forall W \in \mathbb{R}^{n \times m}$, which allows us to show $\mathcal{O}\left(\frac{1}{t}\right)$ convergence of the loss function. We refer readers to Section 4.3 for details.*

### 4.3 CONVERGENCE RESULTS FOR CLASSIFICATION TASKS

As we discussed in Remark 1, the loss functions used in classification tasks generally do not satisfy our assumption **A1** for $f$. Suppose instead we have the following assumption for $f$.

**Assumption 2.** *$f$ satisfies **(A1')** $\|\nabla f(W)\|_F \ge \gamma(f(W) - f^*), \forall W \in \mathbb{R}^{n \times m}$.*

Then we can show $\mathcal{O}\left(\frac{1}{t}\right)$ convergence of the loss function, as stated below.

**Theorem 4.** *Given initialization $\{W_l(0)\}_{l=1}^L$ such that $\lambda_{min}(\mathcal{T}_{\{W_l(t)\}_{l=1}^L}) \ge \alpha, \ \forall t \ge 0$, and $f$ satisfying **(A1')**, then*

$$\mathcal{L}(t) - \mathcal{L}^* \le \frac{\mathcal{L}(0) - \mathcal{L}^*}{(\mathcal{L}(0) - \mathcal{L}^*)\alpha\gamma^2 t + 1}. \tag{24}$$

We refer readers to Appendix B for the proof. The lower bound on $\lambda_{min}(\mathcal{T}_{\{W_l(t)\}_{l=1}^{L}})$ can be obtained for different networks by our results in previous sections. The exponential loss satisfies **A1´** (see Appendix D.2)and is studied in Gunasekar et al. (2017); Yun et al. (2020) for diagonal linear networks.

## 5 CONCLUSION AND DISCUSSION

In this paper, we study the convergence of gradient flow on multi-layer linear models with a loss of the form $f(W_1 W_2 \cdots W_L)$, where $f$ satisfies the gradient dominance property. We show that with proper initialization, the loss converges to its global minimum exponentially. Moreover, we derive a lower bound on the convergence rate that depends on two trajectory-specific quantities: the imbalance matrices, which measure the difference between the weights of adjacent layers, and the least singular value of the weight product $W = W_1 W_2 \cdots W_L$. Our analysis applies to various types of multi-layer linear networks, and our assumptions on $f$ are general enough to include loss functions used for both regression and classification tasks. Future directions include extending our results to analyzing gradient descent algorithms as well as to nonlinear networks.

**Convergence of gradient descent**: Exponential convergence of the gradient flow often suggests a linear rate of convergence of gradient descent when the step size is sufficiently small, and Elkabetz & Cohen (2021) formally establishe such a relation. Indeed, Arora et al. (2018a) shows linear rate of convergence of gradient descent on multi-layer linear networks under balanced initialization. A natural future direction is to translate the convergence results under imbalanced initialization for gradient flow to the convergence of gradient descent with a small step size.

**Nonlinear networks**: While the crucial ingredient of our analysis, invariance of weight imbalance, no longer holds in the presence of nonlinearities such as ReLU activations, Du et al. (2018) shows the diagonal entries of the imbalance are preserved, and Le & Jegelka (2022) shows a stronger version of such invariance given additional assumptions on the training trajectory. Therefore, the weight imbalance could still be used to understand the training of nonlinear networks.

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

## A  CONTROLLING PRODUCT WITH MARGIN

Most of our results regarding the lower bound on $\lambda_{\min} \mathcal{T}_{\{W_l\}_{l=1}^L}$ are given as a value that depends on 1) the imbalance of the weights; 2) the minimum singular value of the product $W = \prod_{l=1}^L$. The former is time-invariant, thus is determined at initialization. As we discussed in Section 3, we require the notion of margin to lower bound $\sigma_{\min}(W(t))$ for the entire training trajectory.

The following Lemma that will be used in subsequent proofs.

**Lemma A.1.** *If $f$ satisfies A2, then the gradient flow dynamics* (3) *satisfies*

$$\sigma_{\min}\left(W(t)\right) \geq \sigma_{\min}\left(W^*\right) - \sqrt{\frac{K}{\mu}} \|W(0) - W^*\|_F, \forall t \geq 0$$

*where $W(t) = \prod_{l=1}^L W_l(t)$ and $W^*$ is the unique minimizer of $f$.*

*Proof.* From Polyak (1987), we know if $f$ is $\mu$-strongly convex, then it has unique minimizer $W^*$ and

$$f(W) - f^* \geq \frac{\mu}{2} \|W - W^*\|_F^2.$$

Additionally, if $f$ is $K$-smooth, then

$$f(W) - f^* \leq \frac{K}{2} \|W - W^*\|_F^2.$$

This suggests that for any $t \geq 0$,

$$\frac{K}{2} \|W(t) - W^*\|_F^2 \geq \mathcal{L}(t) - \mathcal{L}^* \geq \frac{\mu}{2} \|W - W^*\|_F^2.$$

Therefore we have the following

$$
\begin{aligned}
\sigma_{\min}\left(W(t)\right) =\ & \sigma_{\min}\left(W(t) - W^* + W^*\right) \\
\text{(Weyl's inequality (Horn \& Johnson, 2012, 7.3.P16))} \geq\ & \sigma_{\min}(W^*) - \|W(t) - W^*\|_2 \\
\geq\ & \sigma_{\min}(W^*) - \|W(t) - W^*\|_F \\
\text{($f$ is $\mu$-strongly convex)} \geq\ & \sigma_{\min}(W^*) - \sqrt{\frac{2}{\mu}(\mathcal{L}(t) - \mathcal{L}^*)} \\
\text{($\mathcal{L}(t)$ non-decreasing under (3))} \geq\ & \sigma_{\min}(W^*) - \sqrt{\frac{2}{\mu}(\mathcal{L}(0) - \mathcal{L}^*)} \\
\text{($f$ is $K$-smooth)} \geq\ & \sigma_{\min}(W^*) - \sqrt{\frac{K}{\mu}\|W(0) - W^*\|_F^2} \\
=\ & \sigma_{\min}\left(W^*\right) - \sqrt{\frac{K}{\mu}}\|W(0) - W^*\|_F.
\end{aligned}
$$

$\square$

Lemma A.1 directly suggests

$$\sigma_{\min}(W(t)) \geq \left[\sigma_{\min}\left(W^*\right) - \sqrt{\frac{K}{\mu}}\|W(0) - W^*\|_F\right]_+ := margin,$$

and the margin is positive when the initial product $W(0)$ is sufficiently close to the optimal $W^*$.

# B  CONVERGENCE ANALYSIS FOR CLASSIFICATION LOSSES

In this section, we consider $f$ that satisfies, instead of **A1**, the following

**Assumption 3.** $f$ satisfies (**A1´**) the Łojasiewicz inequality-like condition
$$\|\nabla f(W)\|_F \geq \gamma(f(W) - f^*), \forall W \in \mathbb{R}^{n \times m} \,.$$

**Theorem 4** (Restated). *Given initialization $\{W_l(0)\}_{l=1}^L$ such that*
$$\lambda_{min} \mathcal{T}_{\{W_l(t)\}_{l=1}^L} \geq \alpha, \ \forall t \geq 0 \,,$$

*and $f$ satisfying (**A1´**), then*
$$\mathcal{L}(t) - \mathcal{L}^* \leq \frac{\mathcal{L}(0) - \mathcal{L}^*}{(\mathcal{L}(0) - \mathcal{L}^*)\alpha\gamma^2 t + 1} \,.$$

*Proof.* When $f$ satisfies (**A1´**), then (5) becomes
$$\dot{\mathcal{L}} = -\left\langle \mathcal{T}_{\{W_l\}_{l=1}^L} \nabla f(W), \nabla f(W) \right\rangle_F$$
$$\leq -\lambda_{min}\left(\mathcal{T}_{\{W_l\}_{l=1}^L}\right)\|\nabla f(W)\|_F^2$$
$$(\mathbf{A1'}) \leq -\lambda_{min}\left(\mathcal{T}_{\{W_l\}_{l=1}^L}\right)\gamma^2(f(W) - f^*)^2 = -\lambda_{min}\left(\mathcal{T}_{\{W_l\}_{l=1}^L}\right)\gamma^2(\mathcal{L} - \mathcal{L}^*)^2 \,.$$

This shows
$$-\frac{1}{(\mathcal{L} - \mathcal{L}^*)^2}\frac{d}{dt}(\mathcal{L} - \mathcal{L}^*) \geq \lambda_{min}\left(\mathcal{T}_{\{W_l\}_{l=1}^L}\right)\gamma^2 \geq \alpha\gamma^2 \,.$$

Take integral $\int dt$ on both sides, we have for any $t \geq 0$,
$$\left.\frac{1}{\mathcal{L} - \mathcal{L}^*}\right|_0^t \geq \alpha\gamma^2 t \,,$$

which is
$$\mathcal{L}(t) - \mathcal{L}^* \leq \frac{\mathcal{L}(0) - \mathcal{L}^*}{(\mathcal{L}(0) - \mathcal{L}^*)\alpha\gamma^2 t + 1} \,.$$
$\square$

Following similar argument as in Yun et al. (2020), we can show that exponential loss on linearly separable data satisfies **A1´**.

**Claim.** *Let $f(w) = \sum_{i=1}^N \exp\left(-y_i \cdot (x_i^T w)\right)$, if there exists $z \in \mathbb{S}^{n-1}$ and $\gamma > 0$ such that*
$$y_i(x_i^T z) \geq \gamma, \forall i = 1, \cdots, N \,,$$

*then*
$$\|\nabla f(w)\|_F \geq \gamma f(w), \forall w \in \mathbb{R}^n \,.$$

*Proof.* Using the linear separability, we have
$$\|\nabla f(w)\|_F^2 = \left\|\sum_{i=1}^N \exp\left(-y_i \cdot (x_i^T w)\right) y_i x_i\right\|_F^2$$
$$(\text{Cauchy-Schwarz inequality}) \geq \left|\left\langle z, \sum_{i=1}^N \exp\left(-y_i \cdot (x_i^T w)\right) y_i x_i\right\rangle\right|^2$$
$$\geq \left|\sum_{i=1}^N \exp\left(-y_i \cdot (x_i^T w)\right)\gamma\right|^2 = |f(w)\gamma|^2 \,,$$

as desired. $\square$

Therefore, our convergence results applies to classification tasks with exponential loss.

## C  PROOFS IN SECTION 2

First we prove the expression for $\dot{\mathcal{L}}$ in Lemma 1

**Lemma 1** (Restated). *Under continuous dynamics in* (3), *we have*

$$\dot{\mathcal{L}} = -\|\nabla L\left(\{W_l\}_{l=1}^L\right)\|_F^2 = -\left\langle \mathcal{T}_{\{W_l\}_{l=1}^L} \nabla f(W), \nabla f(W)\right\rangle_F,$$

*where $W = \prod_{l=1}^L W_i$, and $\mathcal{T}_{\{W_l\}_{l=1}^L}$ is a positive semi-definite linear operator on $\mathbb{R}^{n\times m}$ with*

$$\mathcal{T}_{\{W_l\}_{l=1}^L} E = \sum_{l=1}^L \left(\prod_{i=1}^{l-1} W_i\right)\left(\prod_{i=1}^{l-1} W_i\right)^T E \left(\prod_{i=l+1}^{L+1} W_i\right)^T \left(\prod_{i=l+1}^{L+1} W_i\right), W_0 = I_n, W_{L+1} = I_m.$$

*Proof.* The gradient flow dynamics (3) satisfies

$$\frac{d}{dt} W_l = -\frac{\partial}{\partial W_l}\mathcal{L}\left(\{W_l\}_{l=1}^L\right) = -\left(\prod_{i=1}^{l-1} W_i\right)^T \nabla f(W)\left(\prod_{i=l+1}^{L+1} W_i\right)^T, \qquad (\text{C.1})$$

where $W = \prod_{l=1}^L W_i$ and $W_0 = I_n, W_{L+1} = I_m$.

Therefore

$$\dot{\mathcal{L}} = \sum_{l=1}^L \left\langle \frac{\partial}{\partial W_l}\mathcal{L}\left(\{W_l\}_{l=1}^L\right), \frac{d}{dt} W_l\right\rangle_F$$

$$= -\sum_{l=1}^L \left\|\frac{\partial}{\partial W_l}\mathcal{L}\left(\{W_l\}_{l=1}^L\right)\right\|_F^2$$

$$= -\sum_{l=1}^L \left\langle \left(\prod_{i=1}^{l-1} W_i\right)^T \nabla f(W)\left(\prod_{i=l+1}^{L+1} W_i\right)^T, \left(\prod_{i=1}^{l-1} W_i\right)^T \nabla f(W)\left(\prod_{i=l+1}^{L+1} W_i\right)^T\right\rangle_F$$

$$= -\sum_{l=1}^L \left\langle \left(\prod_{i=1}^{l-1} W_i\right)\left(\prod_{i=1}^{l-1} W_i\right)^T \nabla f(W)\left(\prod_{i=l+1}^{L+1} W_i\right)^T \left(\prod_{i=l+1}^{L+1} W_i\right), \nabla f(W)\right\rangle_F$$

$$= -\left\langle \sum_{l=1}^L \left(\prod_{i=1}^{l-1} W_i\right)\left(\prod_{i=1}^{l-1} W_i\right)^T \nabla f(W)\left(\prod_{i=l+1}^{L+1} W_i\right)^T \left(\prod_{i=l+1}^{L+1} W_i\right), \nabla f(W)\right\rangle_F$$

$$= -\left\langle \mathcal{T}_{\{W_l\}_{l=1}^L} \nabla f(W), \nabla f(W)\right\rangle_F.$$

$\square$

Next, we prove that the imbalance matrices are time-invariant

**Lemma 2** (Restated). *Under continuous dynamics* (3), *we have $\dot{D}_l(t) = 0, \forall t \geq 0, l = 1, \cdots, L-1$.*

*Proof.* Each imbalance matrix is defined as

$$D_l = W_l^T W_l - W_{l+1} W_{l+1}^T, \ l = 1, \cdots, L-1$$

We only need to check that $\frac{d}{dt}\left(W_l^T W_l\right)$ and $\frac{d}{dt}\left(W_{l+1} W_{l+1}^T\right)$ are identical.

From the following derivation, for $l = 1, \cdots, L - 1$,

$$\frac{d}{dt}\left(W_l^T W_l\right)$$
$$= \dot{W}_l^T W_l + W_l^T \dot{W}_l$$
$$= -\left(\prod_{i=l+1}^{L+1} W_i\right) \nabla^T f(W) \left(\prod_{i=1}^{l-1} W_i\right) W_l - W_l^T \left(\prod_{i=1}^{l-1} W_i\right)^T \nabla f(W) \left(\prod_{i=l+1}^{L+1} W_i\right)^T$$
$$= -\left(\prod_{i=l+1}^{L+1} W_i\right) \nabla^T f(W) \left(\prod_{i=1}^{l} W_i\right) - \left(\prod_{i=1}^{l} W_i\right)^T \nabla f(W) \left(\prod_{i=l+1}^{L+1} W_i\right)^T,$$

$$\frac{d}{dt}\left(W_{l+1} W_{l+1}^T\right)$$
$$= \dot{W}_{l+1} W_{l+1}^T + W_{l+1} \dot{W}_{l+1}^T$$
$$= -\left(\prod_{i=1}^{l} W_i\right)^T \nabla f(W) \left(\prod_{i=l+2}^{L+1} W_i\right)^T W_{l+1}^T - W_{l+1} \left(\prod_{i=l+2}^{L+1} W_i\right) \nabla^T f(W) \left(\prod_{i=1}^{l} W_i\right)$$
$$= -\left(\prod_{i=1}^{l} W_i\right)^T \nabla f(W) \left(\prod_{i=l+1}^{L+1} W_i\right)^T - \left(\prod_{i=l+1}^{L+1} W_i\right) \nabla^T f(W) \left(\prod_{i=1}^{l} W_i\right)$$

we know $\frac{d}{dt}\left(W_l^T W_l\right) = \frac{d}{dt}\left(W_{l+1} W_{l+1}^T\right)$, therefore $\dot{D}_l(t) = 0, l = 1, \cdots, L - 1$  □

## D    Linear Models Related to Scalar Dynamics

### D.1    Spectral Initialization under $l_2$ loss

The spectral initialization Saxe et al. (2014); Gidel et al. (2019); Tarmoun et al. (2021) considers the following:

Suppose $f = \frac{1}{2}\|Y - XW\|_F^2$ and we have overparametrized model $W = \prod_{l=1}^L W_l$. Additionally, we assume $Y \in \mathbb{R}^{N \times m}, X \in \mathbb{R}^{N \times n}$ $(n \geq m)$ are *co-diagonalizable*, i.e. there exist $P \in \mathbb{R}^{N \times n}$ with $P^T P = I_n$ and $Q \in \mathcal{O}(m), R \in \mathcal{O}(n)$ such that we can write the SVDs of $Y, X$ as $Y = P \begin{bmatrix} \Sigma_Y & 0 \\ 0 & 0 \end{bmatrix} \begin{bmatrix} Q \\ 0 \end{bmatrix} := P \tilde{\Sigma}_Y \tilde{Q}$ and $X = P \Sigma_X R^T$.

**Remark 2.** *In Section 4, we discussed the case $f = \frac{1}{2}\|Y - W\|_F^2$, which is essentially considering the aforementioned setting with $N = n$ and $X = I_n$.*

Given any set of weights $\{W_l\}_{l=1}^L$ such that
$$W_1 = R\Sigma_1 V_1^T, \quad W_l = V_{l-1}\Sigma_l V_l^T, \ l = 2, \cdots, L-1, \quad W_L = V_{L-1}\Sigma_L \tilde{Q},$$
where $\Sigma_l, l = 1, \cdots, L$ are diagonal matrices and $V_l \in \mathbb{R}^{n \times h_l}, l = 1, \cdots, L-1$ with $V_l^T V_l = I_{h_l}$. The gradient flow dynamics requires

$$
\begin{aligned}
\dot{W}_1 &= -\frac{\partial \mathcal{L}}{\partial W_1} \\
&= -X^T(Y - XW)W_L^T W_{L-1}^T \cdots W_2^T \\
&= -R\Sigma_X P^T \cdot (P\tilde{\Sigma}_Y \tilde{Q} - P\Sigma_X R^T \cdot R \prod_{l=1}^L \Sigma_l \tilde{Q}) \cdot \tilde{Q}^T \Sigma_L V_{L-1} \cdot V_{L-1}\Sigma_{L-1} V_{L-2}^T \cdots V_2 \Sigma_2 V_1^T \\
&= -R\left(\Sigma_X\left(\Sigma_Y - \Sigma_X \prod_{l=1}^L \Sigma_l\right)\tilde{Q}\tilde{Q}^T \prod_{l=2}^L \Sigma_l\right)V_1^T \\
&= -R\left(\Sigma_X\left(\Sigma_Y - \Sigma_X \prod_{l=1}^L \Sigma_l\right)\begin{bmatrix} I_m & 0 \\ 0 & 0 \end{bmatrix} \prod_{l=2}^L \Sigma_l\right)V_1^T,
\end{aligned}
$$

which shows that the singular space $R, V_1$ for $W_1$ do not change under the gradient flow, and the singular values $\sigma_{i,1}$ of $W_1$ satisfies

$$\dot{\sigma}_{i,1} = \left(\sigma_{i,y} - \sigma_{i,x} \prod_{l=1}^L \sigma_{i,l}\right) \sigma_{i,x} \prod_{l=2}^L \sigma_{i,l}, i = 1, \cdots, m,$$

and $\dot{\sigma}_{i,1} = 0, i = m+1, \cdots, n$.

Similarly, we can show that

$$\dot{W}_l = V_{l-1}\left(\Sigma_X\left(\Sigma_Y - \Sigma_X \prod_{i=1}^L \Sigma_i\right)\begin{bmatrix} I_m & 0 \\ 0 & 0 \end{bmatrix} \prod_{i \neq l}^L \Sigma_i\right)V_l^T, l = 2, \cdots, L-1,$$

$$\dot{W}_L = V_{L-1}\left(\Sigma_X\left(\Sigma_Y - \Sigma_X \prod_{i=1}^L \Sigma_i\right)\begin{bmatrix} I_m & 0 \\ 0 & 0 \end{bmatrix} \prod_{i \neq L}^L \Sigma_i\right)\tilde{Q}.$$

Overall, this suggests that the singular space of $\{W_l\}_{l=1}^L$ do not change under the gradient flow, and their singular values satisfies, for $i = 1, \cdots, m$,

$$\dot{\sigma}_{i,l} = \left(\sigma_{i,y} - \sigma_{i,x} \prod_{k=1}^L \sigma_{i,k}\right) \sigma_{i,x} \prod_{k \neq l}^L \sigma_{i,k}, l = 1, \cdots, L.$$

Each dynamic equation is equivalent to the one from gradient flow on $\mathcal{L}_i(\{\sigma_{i,l}\}_{l=1}^L) = \frac{1}{2}\left|\sigma_{i,y} - \sigma_{i,x}\prod_{l=1}^L \sigma_{i,l}\right|^2$. Therefore, under spectral initialization, the dynamics of the weights are decoupled into at most $m$ dynamics discussed in Section 4.2.

## D.2 DIAGONAL LINEAR NETWORKS

The loss function of diagonal linear networks Gunasekar et al. (2017); Yun et al. (2020) is of the form $f(w_1 \odot \cdots \odot w_L)$, we write

$$
\mathcal{L}(\{w_l\}_{l=1}^L) = f(w_1 \odot \cdots \odot w_L) = f(w^{(1)}, \cdots, w^{(n)}) = f\left( \prod_{l=1}^L w_{l,1}, \cdots, \prod_{l=1}^L w_{l,n} \right),
$$

i.e. $f$ takes $n$ variables $w^{(1)}, \cdots, w^{(n)}$ and each variable $w^{(i)}$ is overparametrized into $\prod_{l=1}^L w_{l,i}$. Then we can show that

$$
\begin{aligned}
\dot{\mathcal{L}} &= -\|\nabla_{\{w_l\}_{l=1}^L} \mathcal{L}\|_F^2 \\
&= \sum_{i=1}^n \sum_{l=1}^L \left| \frac{\partial \mathcal{L}}{\partial w_{l,i}} \right|^2 \\
&= \sum_{i=1}^n \sum_{l=1}^L \left| \frac{\partial f}{\partial w^{(i)}} \right|^2 \left| \frac{\partial w^{(i)}}{\partial w_{l,i}} \right|^2 \\
&= \sum_{i=1}^n \left| \frac{\partial f}{\partial w^{(i)}} \right|^2 \sum_{l=1}^L \left| \frac{\partial w^{(i)}}{\partial w_{l,i}} \right|^2 \\
&= \sum_{i=1}^n \left| \frac{\partial f}{\partial w^{(i)}} \right|^2 \tau_{\{w_{l,i}\}_{l=1}^L} \\
&\leq -\left( \min_{1 \leq i \leq n} \tau_{\{w_{l,i}\}_{l=1}^L} \right) \sum_{i=1}^n \left| \frac{\partial f}{\partial w^{(i)}} \right|^2 \\
(f \text{ satisfies } \mathbf{A1}) &\leq -\left( \min_{1 \leq i \leq n} \tau_{\{w_{l,i}\}_{l=1}^L} \right) \gamma(f - f^*) = -\left( \min_{1 \leq i \leq n} \tau_{\{w_{l,i}\}_{l=1}^L} \right) \gamma(\mathcal{L} - \mathcal{L}^*).
\end{aligned}
$$

Moreover, the imbalances $\{d_l^{(i)} := w_{l,i}^2 - w_{l+1,i}^2\}_{l=1}^{L-1}$ are time-invariant for each $i = 1, \cdots, n$ by Lemma 2. Therefore, we can lower bound each $\tau_{\{w_{l,i}\}_{l=1}^L}$ using the imbalance $\{d_l^{(i)}\}_{l=1}^{L-1}$ as in Proposition 3, from which one obtain the exponential convergence of $\mathcal{L}$.

# E   PROOF FOR TWO-LAYER MODEL

Using Lemma 3, we can prove Theorem 1

**Theorem 1** (Restated). *Let $D$ be the imbalance matrix for $L = 2$. The continuous dynamics in* (3) *satisfy*

$$\mathcal{L}(t) - \mathcal{L}^* \leq \exp\left(-\alpha_2 \gamma t\right)\left(\mathcal{L}(0) - \mathcal{L}^*\right), \forall t \geq 0\,, \tag{E.2}$$

*where*

1. *If $f$ satisfies only **A1**, then $\alpha_2 = \underline{\Delta}$;*

2. *If $f$ satisfies both **A1** and **A2**, then*

$$\alpha_2 = -\Delta_+ + \sqrt{(\Delta_+ + \underline{\Delta})^2 + 4\left(\left[\sigma_n\left(W^*\right) - \sqrt{K/\mu}\|W(0) - W^*\|_F\right]_+\right)^2}$$
$$- \Delta_- + \sqrt{(\Delta_- + \underline{\Delta})^2 + 4\left(\left[\sigma_m\left(W^*\right) - \sqrt{K/\mu}\|W(0) - W^*\|_F\right]_+\right)^2}\,, \tag{E.3}$$

*with $W(0) = \prod_{l=1}^{L} W_l(0)$ and $W^*$ equal to the unique optimizer of $f$.*

*Proof.* As shown in (5) in Section 2. We have

$$\frac{d}{dt}(\mathcal{L}(t) - \mathcal{L}^*) \leq -\lambda_{\min}\mathcal{T}_{\{W_1(t), W_2(t)\}}\gamma(\mathcal{L}(t) - \mathcal{L}^*)\,.$$

Consider any $\{W_1(t), W_2(t)\}$ on the trajectory, we have, by Lemma 3,

$$\lambda_{\min}\mathcal{T}_{\{W_1(t), W_2(t)\}} \stackrel{\text{Lemma 3}}{\geq} \frac{1}{2}\left(-\Delta_+ + \sqrt{(\Delta_+ + \underline{\Delta})^2 + 4\sigma_n^2\left(W(t)\right)}\right.$$
$$\left. -\Delta_- + \sqrt{(\Delta_- + \underline{\Delta})^2 + 4\sigma_m^2\left(W(t)\right)}\right)$$
$$\geq \frac{1}{2}\left(-\Delta_+ + \sqrt{(\Delta_+ + \underline{\Delta})^2} - \Delta_- + \sqrt{(\Delta_- + \underline{\Delta})^2}\right) = \underline{\Delta} := \alpha_2\,.$$

**When $f$ also satisfies A2**: we need to prove

$$\sigma_n\left(W(t)\right) \geq \left[\sigma_n\left(W^*\right) - \sqrt{K/\mu}\|W(0) - W^*\|_F\right]_+\,, \tag{E.4}$$

$$\sigma_m\left(W(t)\right) \geq \left[\sigma_m\left(W^*\right) - \sqrt{K/\mu}\|W(0) - W^*\|_F\right]_+\,. \tag{E.5}$$

*When $n = m$*, both inequalities are equivalent to

$$\sigma_{\min}(W(t)) \geq \left[\sigma_{\min}(W^*) - \sqrt{K/\mu}\|W(0) - W^*\|_F\right]_+\,,$$

which is true by Lemma A.1.

*When $n \neq m$*, one of the two inequalities become trivial. For example, if $n > m$, then (E.4) is trivially $0 \geq 0$, and (E.5) is equivalent to

$$\sigma_{\min}(W(t)) \geq \left[\sigma_{\min}(W^*) - \sqrt{K/\mu}\|W(0) - W^*\|_F\right]_+\,,$$

which is true by Lemma A.1.

Overall, we have

$$\lambda_{min} \mathcal{T}_{\{W_1(t), W_2(t)\}}$$

$$\stackrel{\text{Lemma 3}}{\geq} \frac{1}{2} \left( -\Delta_+ + \sqrt{(\Delta_+ + \underline{\Delta})^2 + 4\sigma_n^2 \left( W(t) \right)} \right.$$

$$\left. -\Delta_- + \sqrt{(\Delta_- + \underline{\Delta})^2 + 4\sigma_m^2 \left( W(t) \right)} \right)$$

$$\geq \frac{1}{2} \left( -\Delta_+ + \sqrt{(\Delta_+ + \underline{\Delta})^2 + 4 \left( \left[ \sigma_n \left( W^* \right) - \sqrt{K/\mu} \| W(0) - W^* \|_F \right]_+ \right)^2} \right.$$

$$\left. -\Delta_- + \sqrt{(\Delta_- + \underline{\Delta})^2 + 4 \left( \left[ \sigma_m \left( W^* \right) - \sqrt{K/\mu} \| W(0) - W^* \|_F \right]_+ \right)^2} \right)$$

$$:= \alpha_2 \,.$$

Either case, we have $\frac{d}{dt} (\mathcal{L}(t) - \mathcal{L}^*) \leq -\alpha_2 \gamma (\mathcal{L}(t) - \mathcal{L}^*)$, and by Grönwall's inequality, we have

$$\mathcal{L}(t) - \mathcal{L}^* \leq \exp(-\alpha_2 \gamma t)(\mathcal{L}(0) - \mathcal{L}^*) \,.$$

$\square$

# F    PROOFS FOR THREE-LAYER MODEL

In Section F.1, we discuss the proof idea for Theorem 2, then present the proof afterwards. In Section G, we show a simplified bound when the weights can be ordered w.r.t. positive-semidefiniteness.

## F.1    PROOF IDEA

We first discuss the proof idea behind Theorem 2, then provide the complete proof. Consider the case when $n = m = 1$, we use the following notations for the weights $\{w_1^T, W_2, w_3\} \in \mathbb{R}^{1 \times h_1} \times \mathbb{R}^{h_1 \times h_2} \times \mathbb{R}^{h_2 \times 1}$. The quantity we need to lower bound is

$$\lambda_{\min} \mathcal{T}_{\{w_1^T, W_2, w_3\}} = w_1^T W_2 W_2^T w_1 + w_1^T w_1 \cdot w_3^T w_3 + w_3^T W_2^T W_2 w_3$$
$$= \|W_2^T w_1\|^2 + \|w_1\|^2 \|w_3\|^2 + \|W_2 w_3\|^2 \,,$$

where our linear operator $\mathcal{T}_{\{w_1^T, W_2, w_3\}}$ reduces to a scalar. The remaining thing to do is to find

$$\min_{w_1^T, W_2, w_3} \|W_2^T w_1\|^2 + \|w_1\|^2 \|w_3\|^2 + \|W_2 w_3\|^2 \tag{F.6}$$
$$s.t. \quad W_2 W_2^T - w_1 w_1^T = D_{21}$$
$$W_2^T W_2 - w_3 w_3^T = D_{23}$$

i.e., we try to find the best lower bound on $\lambda_{\min} \mathcal{T}_{\{w_1^T, W_2, w_3\}}$ given the fact that the weights have to satisfies the imbalance constraints from $D_{21}, D_{23}$, and $\lambda_{\min} \mathcal{T}_{\{w_1^T, W_2, w_3\}}$ is related to the norm of some weights $\|w_1\|, \|w_3\|$ and the "alignment" between weights $\|W_2^T w_1\|, \|W_2 w_3\|$.

The general idea of the proof is to lower bound each term $\|W_2^T w_1\|^2, \|w_1\|^2, \|w_3\|^2, \|W_2 w_3\|^2$ individually given the imbalance constraints, then show the existence of some $\{w_1^T, W_2, w_3\}$ that attains the lower bound simultaneously. The following discussion is most for lower bounding $\|w_1\|, \|W_2^T w_1\|$ but the same argument holds for lower bounding other quantities.

Understanding what can be chosen to be the spectrum of $W_2 W_2^T (W_2^T W_2)$ is the key to derive an lower bound, and the imbalance constraints implicitly limit such choices. To see this, notice that $W_2 W_2^T - w_1 w_1^T = D_{21}$ suggests an eigenvalue *interlacing relation* (Horn & Johnson, 2012, Corollary 4.39) between $W_2 W_2^T$ and $D_{21}$, i.e.

$$\lambda_{h_1}(D_{21}) \le \lambda_{h_1}(W_2 W_2^T) \le \lambda_{h_1-1}(D_{21}) \le \cdots \le \lambda_2(W_2 W_2^T) \le \lambda_1(D_{21}) \le \lambda_1(W_2 W_2^T) \,.$$

Therefore, any choice of $\{\lambda_i(W_2 W_2^T)\}_{i=1}^{h_1}$ must satisfy the interlacing relation with $\{\lambda_i(D_{21})\}_{i=1}^{h_1}$. Similarly, $\{\lambda_i(W_2^T W_2)\}_{i=1}^{h_2}$ must satisfy the interlacing relation with $\{\lambda_i(D_{23})\}_{i=1}^{h_2}$. Moreover, $\{\lambda_i(W_2 W_2^T)\}_{i=1}^{h_1}$ and $\{\lambda_i(W_2^T W_2)\}_{i=1}^{h_2}$ agree on non-zero eigenvalues. In short, an appropriate choice of the spectrum of $W_2 W_2^T (W_2^T W_2)$ needs to respect the interlacing relation with the eigenvalues of $D_{21}$ and $D_{23}$.

The following matrix is defined

$$\bar{D}_{h_1} := \mathrm{diag}\{\max\{\lambda_i(D_{21}), \lambda_i(D_{23}), 0\}\}_{i=1}^{h_1}$$

to be the "minimum" choice of the spectrum of $W_2 W_2^T (W_2^T W_2)$ in the sense that any valid choice of $\{\lambda_i(W_2 W_2^T)\}_{i=1}^{h_1}$ must satisfies

$$\lambda_i(W_2 W_2^T) \ge \lambda_i(\bar{D}_{h_1}) \ge \lambda_i(D_{21}) \,, i = 1, \cdots, h_1 \,.$$

That is, the spectrum of $\bar{D}_{h_1}$ "lies between" the one of $W_2 W_2^T$ and of $D_{21}$. Now we check the imbalance constraint again $W_2 W_2^T - w_1 w_1^T = D_{21}$, it shows that: *using a rank-one update $w_1 w_1^T$, one obtain the spectrum of $D_{21}$ starting from the spectrum of $W_2 W_2^T$*, and more importantly, we require the norm $\|w_1\|^2$ to be (taking the trace on the imbalance equation)

$$\mathrm{tr}(W_2 W_2^T) - \|w_1\|^2 = \mathrm{tr}(D_{21}) \quad \Rightarrow \quad \|w_1\|^2 = \mathrm{tr}(W_2 W_2^T) - \mathrm{tr}(D_{21}) \,.$$

Now since $\bar{D}_{h_1}$ "lies inbetween", we have

$$\|w_1\|^2 = \mathrm{tr}(W_2 W_2^T) - \mathrm{tr}(D_{21})$$
$$= (\text{changes from } \lambda_i(W_2 W_2^T) \text{ to } \lambda_i(D_{21}))$$
$$= (\text{changes from } \lambda_i(W_2 W_2^T) \text{ to } \lambda_i(\bar{D}_{h_1})) + (\text{changes from } \lambda_i(\bar{D}_{h_1}) \text{ to } \lambda_i(D_{21}))$$
$$\ge (\text{changes from } \lambda_i(\bar{D}_{h_1}) \text{ to } \lambda_i(D_{21})) = \mathrm{tr}(\bar{D}_{h_1}) - \mathrm{tr}(D_{21}) \,,$$

which is a lower bound on $\|w_1\|^2$. It is exactly the $\Delta_{21}$ in Theorem 2 (It takes more complicated form when $n > 1$).

A lower bound on $\|W_2^T w_1\|^2$ requires carefully exam the changes from the spectrum of $\bar{D}_{h_1}$ to the one of $D_{21}$. If $\lambda_{h_1}(D_{21}) < 0$, then "changes from $\lambda_i(\bar{D})$ to $\lambda_i(D_{21})$" has two parts

1. (changes from $\lambda_i(\bar{D})$ to $[\lambda_i(D_{21})]_+$) through the part where $w_1$ is "aligned" with $W_2^T$,
2. (changes from 0 to $\lambda_{h_1}(D_{21})$) through the part where $w_1$ is "orthogonal" to $W_2^T$.

Only the former contributes to $\|W_2^T w_1\|^2$ hence we need the expression $\Delta_{21}^{(2)} + \Delta_{21}^2$, which excludes the latter part. Using similar argument we can lower bound $\|w_3\|^2, \|W_2 w_3\|^2$. Lastly, the existence of $\{w_1^T, W_2, w_3\}$ that attains the lower bound is from the fact that $\bar{D}_{h_1}(\bar{D}_{h_2})$ is a valid choice for the spectrum of $W_2 W_2^T (W_2^T W_2)$.

The complete proof of the Theorem 2 follows the same idea but with a generalized notion of eigenvalue interlacing, and some related novel eigenvalue bounds.

## F.2 PROOF OF THEOREM 2

Theorem 2 is the direct consequence of the following two results.

**Lemma F.1.** *Given any set of weights* $\{W_1, W_2, W_3\} \in \mathbb{R}^{n \times h_1} \times \mathbb{R}^{h_1 \times h_2} \times \mathbb{R}^{h_2 \times m}$, *we have*

$$\lambda_{min}\mathcal{T}_{\{W_1,W_2,W_3\}} \geq \lambda_n(W_1 W_2 W_2^T W_1^T) + \lambda_n(W_1 W_1^T)\lambda_m(W_3^T W_3) + \lambda_m(W_3^T W_2^T W_2 W_3).$$

(Note that $\lambda_{min}\mathcal{T}_{\{W_1,W_2,W_3\}}$ does not have a closed-form expression. One can only work with its lower bound $\lambda_n(W_1 W_2 W_2^T W_1^T) + \lambda_n(W_1 W_1^T)\lambda_m(W_3^T W_3) + \lambda_m(W_3^T W_2^T W_2 W_3)$.)

**Theorem F.2.** *Given imbalance matrices pair* $(D_{21}, D_{23}) \in \mathbb{R}^{h_1 \times h_1} \times \mathbb{R}^{h_2 \times h_2}$, *then the optimal value of*

$$\min_{W_1,W_2,W_3} 2\left(\lambda_n(W_1 W_2 W_2^T W_1^T) + \lambda_n(W_1 W_1^T)\lambda_m(W_3^T W_3) + \lambda_m(W_3^T W_2^T W_2 W_3)\right)$$

$$s.t. \quad W_2 W_2^T - W_1^T W_1 = D_{21}$$

$$W_2^T W_2 - W_3 W_3^T = D_{23}$$

*is*

$$\Delta^*(D_{21}, D_{23}) = \Delta_{21}^{(2)} + \Delta_{21}^2 + 2\Delta_{21}\Delta_{23} + \Delta_{23}^{(2)} + \Delta_{23}^2.$$

Combining those two results gets $\lambda_{min}\mathcal{T}_{\{W_1,W_2,W_3\}} \geq \Delta^*(D_{21}, D_{23})/2$, as stated in Theorem 2.

The Lemma F.1 is intuitive and easy to prove:

*Proof of Lemma F.1.* Notice that $\mathcal{T}_{\{W_1,W_2,W_3\}}$ is the summation of three positive semi-definite linear operators on $\mathbb{R}^{n \times m}$, i.e.

$$\mathcal{T}_{\{W_1,W_2,W_3\}} = \mathcal{T}_{12} + \mathcal{T}_{13} + \mathcal{T}_{23},$$

where

$$\mathcal{T}_{12}E = W_1 W_2 W_2^T W_1^T E, \quad \mathcal{T}_{13}E = W_1 W_1^T E W_3^T W_3, \quad \mathcal{T}_{23}E = E W_3^T W_2^T W_2 W_3,$$

and $\lambda_{min}\mathcal{T}_{12} = \lambda_n(W_1 W_2 W_2^T W_1^T)$, $\lambda_{min}\mathcal{T}_{13} = \lambda_n(W_1 W_1^T)\lambda_m(W_3^T W_3)$, $\lambda_{min}\mathcal{T}_{23} = \lambda_m(W_3^T W_2^T W_2 W_3)$.

Therefore, let $E_{min}$ with $\|E_{min}\|_F = 1$ be the eigenmatrix associated with $\lambda_{min}\mathcal{T}_{\{W_1,W_2,W_3\}}$, we have

$$\begin{aligned}
\lambda_{min}\mathcal{T}_{\{W_1,W_2,W_3\}} &= \langle \mathcal{T}_{\{W_1,W_2,W_3\}}, E_{min}\rangle_F \\
&= \langle \mathcal{T}_{12}, E_{min}\rangle_F + \langle \mathcal{T}_{13}, E_{min}\rangle_F + \langle \mathcal{T}_{23}, E_{min}\rangle_F \\
&\geq \lambda_{min}\mathcal{T}_{12} + \lambda_{min}\mathcal{T}_{13} + \lambda_{min}\mathcal{T}_{23}.
\end{aligned}$$

$\square$

The rest of this section is dedicated to prove Theorem F.2

We will first state a few Lemmas that will be used in the proof, then show the proof for Theorem F.2, and present the long proofs for the auxiliary Lemmas in the end.

### F.3 AUXILIARY LEMMAS

The main ingredient used in proving Theorem F.2 is the notion of $r$-interlacing relation between the spectrum of two matrices, which is a natural generalization of the interlacing relation as seen in classical Cauchy Interlacing Theorem (Horn & Johnson, 2012, Theorem 4.3.17).

**Definition 4.** *Given real symmetric matrices $A, B$ of order $n$, write $A \succeq_r B$, if*

$$\lambda_{i+r}(A) \leq \lambda_i(B) \leq \lambda_i(A), \forall i$$

*where $\lambda_j(\cdot) = +\infty, j \leq 0$ and $\lambda_j(\cdot) = -\infty, j > n$. The case $r = 1$ gives the interlacing relation.*

**Claim.** *We only need to check*

$$\lambda_{i+r}(A) \leq \lambda_i(B) \leq \lambda_i(A), \forall i \in [n],$$

*for showing $A \succeq_r B$.*

*Proof.* Any inequality regarding index outside $[n]$ is trivial. $\qquad\square$

The following Lemma is a direct concequence of Weyl's inequality (Horn & Johnson, 2012, Theorem 4.3.1), and stated as a special case of (Horn & Johnson, 2012, Corollary 4.3.3)

**Lemma F.3.** *Given real symmetric matrices $A, B$ of order $n$, if $A - B$ is positive semi-definite and $\mathrm{rank}(A - B) \leq r$, then $A \succeq_r B$*

The converse is also true

**Lemma F.4.** *Given real symmetric matrices $A, B$ of order $n$, if $A \succeq_r B$, then there exists a positive semi-definite matrix $XX^T$ with $\mathrm{rank}(XX^T) \leq r$ and a real orthogonal matrix $V$ such that $A - XX^T = VBV^T$.*

*Proof.* The case $r = 1$ is proved in (Horn & Johnson, 2012, Theorem 4.3.26). The case $r > 1$ is proved in (Wang & Zheng, 2019, Theorem 1.3) by induction. $\qquad\square$

Specifically for our problem, we also need the following ($\bar{D}_{h_1}$ and $\bar{D}_{h_2}$ are defined in Section 4)

**Lemma F.5.** *Given imbalance matrices pair $(D_{21}, D_{23}) \in \mathbb{R}^{h_1 \times h_1} \times \mathbb{R}^{h_2 \times h_2}$, we have $\bar{D}_{h_1} \succeq_n D_{21}$ and $\bar{D}_{h_2} \succeq_m D_{23}$.*

In our analysis, the weights $W_1, W_2, W_3$ are "constrained" by the imbalance $D_{21}, D_{23}$, such constraints leads to some special eigenvalue bounds (The operation $\wedge_r$ was defined in Section 4):

**Lemma F.6.** *Given an positive semi-definite matrix $A$ of order $n$, and $Z \in \mathbb{R}^{r \times n}$ with $r \leq n$, when*

$$A - Z^T Z = B,$$

*we have*

$$\lambda_r(ZZ^T) \geq \mathrm{tr}(A) - \mathrm{tr}(A \wedge_r B),$$

*and*

$$2\lambda_r(ZAZ^T) \geq \mathrm{tr}\left(A^2\right) - \mathrm{tr}\left((A \wedge_r B)^2\right) + (\mathrm{tr}(A) - \mathrm{tr}(A \wedge_r B))^2$$

and this bound is actually tight

**Lemma F.7.** *Given two real symmetric matrices $A, B$ of order $n$, if $A \succeq_r B$ ($r \leq n$), then there exist $Z \in \mathbb{R}^{r \times n}$ and some orthogonal matrix $V \in \mathcal{O}(n)$, such that*

$$A - Z^T Z = VBV^T,$$

*and*

$$\lambda_r(ZZ^T) = \mathrm{tr}(A) - \mathrm{tr}(A \wedge_r B),$$
$$2\lambda_r(ZAZ^T) = \mathrm{tr}\left(A^2\right) - \mathrm{tr}\left((A \wedge_r B)^2\right) + (\mathrm{tr}(A) - \mathrm{tr}(A \wedge_r B))^2.$$

**Remark 3.** *To see how Lemma F.6 is used, let $A = W_2 W_2^T$ and $Z = W_1$, $B = D_{21}$, one obtain a lower bound on $\lambda_r(W_1 W_1^T)$ that depends on the entire spectrum of $W_2 W_2^T$ and $D_{21}$. This bound is strictly better than $\lambda_r(W_2 W_2^T) - \lambda_1(D_{21})$, the one from Weyl's inequality (Horn & Johnson, 2012). This should not be suprising because we have "more information" on $W_2 W_2^T$ and $D_{21}$ (entire spectrum v.s. certain eigenvalue).*

### F.4 PROOF OF THEOREM F.2

With these Lemmas, we are ready to prove Theorem F.2.

*Proof of Theorem F.2.* The proof is presented in two parts: First, we show $\Delta^*(D_{21}, D_{23})$ is a lower bound on the optimal value; Then we construct an optimal solution $(W_1^*, W_2^*, W_3^*)$ that attains $\Delta^*(D_{21}, D_{23})$ as the objective value.

**Showing $\Delta^*(D_{21}, D_{23})$ is a lower bound**: Given any feasible triple $(W_1, W_2, W_3)$, the imbalance equations

$$W_2 W_2^T - W_1^T W_1 = D_{21}, \tag{F.7}$$

$$W_2^T W_2 - W_3 W_3^T = D_{23}, \tag{F.8}$$

implies $W_2 W_2^T \succeq_n D_{21}$ and $W_2^T W_2 \succeq_m D_{23}$ by Lemma F.3. These interlacing relation shows

$$\lambda_i(W_2 W_2^T) \geq \lambda_i(D_{21}), \quad \lambda_i(W_2^T W_2) \geq \lambda_i(D_{23}), \forall i,$$

which is

$$\lambda_i(W_2 W_2^T) = \lambda_i(W_2^T W_2) \geq \max\{\lambda_i(D_{21}), \lambda_i(D_{21}), 0\} = \lambda_i(\bar{D}_{h_1}) \geq 0, \forall i \in [h_1] \tag{F.9}$$

Now by Lemma F.6, imbalance equation (F.7) suggests

$$\lambda_n(W_1 W_1^T) \geq \mathrm{tr}(W_2 W_2^T) - \mathrm{tr}(W_2 W_2^T \wedge_n D_{21}),$$

and

$$2\lambda_n(W_1 W_2 W_2^T W_1^T)$$
$$\geq \mathrm{tr}\left((W_2 W_2^T)^2\right) - \mathrm{tr}\left((W_2 W_2^T \wedge_n D_{21})^2\right) + \left(\mathrm{tr}(W_2 W_2^T) - \mathrm{tr}(W_2 W_2^T \wedge_n D_{21})\right)^2.$$

Notice that

$$\begin{aligned}
\lambda_r(W_1 W_1^T) &\geq \mathrm{tr}(W_2 W_2^T) - \mathrm{tr}(W_2 W_2^T \wedge_n D_{21}) \\
&= \sum_{i=1}^{h_1} \lambda_i(W_2 W_2^T) - \min\{\lambda_i(W_2 W_2^T), \lambda_{i+1-n}(D_{21})\} \\
&= \sum_{i=1}^{h_1} \max\{\lambda_i(W_2 W_2^T) - \lambda_{i+1-n}(D_{21}), 0\} \\
&\geq \sum_{i=1}^{h_1} \max\{\lambda_i(\bar{D}_{h_1}) - \lambda_{i+1-n}(D_{21}), 0\} \\
&= \mathrm{tr}(\bar{D}_{h_1}) - \mathrm{tr}(\bar{D}_{h_1} \wedge_n D_{21}) = \Delta_{21}, \tag{F.10}
\end{aligned}$$

where the inequality holds because (F.9) and the fact that ReLU function $f(x) = \max\{x, 0\}$ is a monotonically non-decreasing function.

Since $\Delta_{21}$ can be viewed as summation of ReLU outputs, it has to be non-negative, then (F.10) also suggests

$$\left(\mathrm{tr}(W_2 W_2^T) - \mathrm{tr}(W_2 W_2^T \wedge_n D_{21})\right)^2 \geq \Delta_{21}^2. \tag{F.11}$$

Next we have

$$\begin{aligned}
&2\lambda_n(W_1 W_2 W_2^T W_1^T) \\
&\geq \mathrm{tr}\left((W_2 W_2^T)^2\right) - \mathrm{tr}\left((W_2 W_2^T \wedge_n D_{21})^2\right) + \left(\mathrm{tr}(W_2 W_2^T) - \mathrm{tr}(W_2 W_2^T \wedge_n D_{21})\right)^2 \\
&\overset{\text{(F.11)}}{\geq} \Delta_{21}^2 + \mathrm{tr}\left((W_2 W_2^T)^2\right) - \mathrm{tr}\left((W_2 W_2^T \wedge_n D_{21})^2\right) \\
&= \Delta_{21}^2 + \sum_{i=1}^{h_1} \lambda_i^2(W_2 W_2^T) - \left(\min\{\lambda_i(W_2 W_2^T), \lambda_{i+1-n}(D_{21})\}\right)^2 \\
&\geq \Delta_{21}^2 + \sum_{i=1}^{h_1} \lambda_i^2(\bar{D}_{h_1}) - \left(\min\{\lambda_i(\bar{D}_{h_1}), \lambda_{i+1-n}(D_{21})\}\right)^2 \\
&= \Delta_{21}^2 + \mathrm{tr}\left(\bar{D}_{h_1}^2\right) - \mathrm{tr}\left((\bar{D}_{h_1} \wedge_n D_{21})^2\right) = \Delta_{21}^2 + \Delta_{21}^{(2)},
\end{aligned}$$

where the last inequality is because (F.9) and the fact that the function

$$g(x) = x^2 - (\min\{x, a\})^2 = \begin{cases} 0, & x \le a \\ x^2 - a^2, & x > a \end{cases},$$

is monotonically non-decreasing on $\mathbb{R}_{\ge 0}$ for any constant $a \in \mathbb{R}$.

At this point, we have shown

$$\lambda_n(W_1 W_1^T) \ge \Delta_{21}, \qquad 2\lambda_n(W_1 W_2 W_2^T W_1^T) \ge \Delta_{21}^2 + \Delta_{21}^{(2)}. \tag{F.12}$$

We can repeat the proofs above with the following replacement

$$W_2 \to W_2^T, W_1 \to W_3^T, D_{21} \to D_{23}, \bar{D}_{h_1} \to \bar{D}_{h_2},$$

and obtain

$$\lambda_m(W_3^T W_3) \ge \Delta_{23}, \qquad 2\lambda_m(W_3^T W_2^T W_2 W_3) \ge \Delta_{23}^2 + \Delta_{23}^{(2)}. \tag{F.13}$$

These inequalities (F.12)(F.13) show that

$$\Delta^*(D_{21}, D_{23}) = \Delta_{21}^{(2)} + \Delta_{21}^2 + 2\Delta_{21}\Delta_{23} + \Delta_{23}^{(2)} + \Delta_{23}^2.$$

is a lower bound on the optimal value of our optimization problem. Now we proceed to show tightness.

**Constructing optimal solution**:

By Lemma F.5, we know $\bar{D}_{h_1} \succeq_n D_{21}$, and by Lemma F.7, there exists $Z_1 \in \mathbb{R}^{n \times h_1}$ and orthogonal $V_1 \in \mathcal{O}(h_1)$ such that

$$\bar{D}_{h_1} - Z_1^T Z_1 = V_1 D_{21} V_1^T, \tag{F.14}$$

and most importantly,

$$\lambda_n(Z_1 Z_1^T) = \Delta_{21}, \qquad 2\lambda_n(Z_1 \bar{D}_{h_1} Z_1^T) = \Delta_{21}^{(2)} + \Delta_{21}^2. \tag{F.15}$$

Similarly, by Lemma Lemma F.5, we know $\bar{D}_{h_2} \succeq_m D_{23}$, and by Lemma F.7, there exists $Z_3 \in \mathbb{R}^{m \times h_2}$ and orthogonal $V_3 \in \mathcal{O}(h_2)$ such that

$$\bar{D}_{h_2} - Z_3^T Z_3 = V_3 D_{23} V_3^T, \tag{F.16}$$

and most importantly,

$$\lambda_m(Z_3 Z_3^T) = \Delta_{23}, \qquad 2\lambda_m \left(Z_3 \bar{D}_{h_2} Z_3^T\right) = \Delta_{23}^{(2)} + \Delta_{23}^2. \tag{F.17}$$

Let

$$W_2^* = \begin{cases} V_1^T \left[ \bar{D}^{\frac{1}{2}} \quad \mathbf{0}_{h_1 \times (h_2 - h_1)} \right] V_3, & h_2 \ge h_1 \\ V_1^T \begin{bmatrix} \bar{D}^{\frac{1}{2}} \\ \mathbf{0}_{(h_1 - h_2) \times h_2} \end{bmatrix} V_3, & h_2 < h_1 \end{cases},$$

where $\bar{D} = \mathrm{diag}\{\max\{\lambda_i(D_{21}), \lambda_i(D_{21}), 0\}\}_{i=1}^{\min\{h_1, h_2\}}$, and

$$W_1^* = Z_1 V_1, \qquad W_3^* = V_3^T Z_3^T,$$

we have

$$W_2^*(W_2^*)^T - (W_1^*)^T W_1^* = V_1^T \bar{D}_{h_1} V_1 - V_1^T Z_1^T Z_1 V_1 = D_{21}$$
$$(W_2^*)^T W_2^* - W_3^*(W_3^*)^T = V_3^T \bar{D}_{h_2} V_3 - V_3^T Z_3 Z_3^T V_3 = D_{23},$$

and

$$\lambda_r(W_1^*(W_1^*)^T) = \lambda_r(Z_1 Z_1^T) = \Delta_{21},$$
$$\lambda_m((W_3^*)^T W_3^*) = \lambda_m(Z_3^T Z_3) = \Delta_{23},$$
$$2\lambda_r(W_1^* W_2^*(W_2^*)^T(W_1^*)^T) = \lambda_r(Z_1 \bar{D}_{h_1} Z_1^T) = \Delta_{21}^{(2)} + \Delta_{21}^2,$$
$$2\lambda_m((W_3^*)^T(W_2^*)^T W_2^* W_3^*) = \lambda_m(Z_3^T \bar{D}_{h_2} Z_3) = \Delta_{23}^{(2)} + \Delta_{23}^2,$$

Therefore the lower bound $\Delta^*(D_{21}, D_{23})$ is tight. $\qquad \square$

F.5 PROOFS FOR AUXILIARY LEMMAS

We finish this section by providing the proofs for auxiliary lemmas we used in the last section.

*Proof of Lemma F.5.* Since $(D_{21}, D_{23})$ is a pair of imbalance matrices, there exists $W_2 W_2^T$, such that

$$W_2 W_2^T \succeq_n D_{21}, W_2^T W_2 \succeq_m D_{23}, \tag{F.18}$$

because at least our weight initialization $W_1(0), W_2(0), W_3(0)$ have to satisfy $W_2(0)W_2(0)^T - W_1^T(0)W_1(0) = D_{21}, W_2^T(0)W_2(0) - W_3(0)W_3^T(0) = D_{23}$.

Therefore, for $0 < i \leq h_1 - n$,

$$\lambda_{i+n}(\bar{D}_{h_1}) = \max\{\lambda_{i+n}(D_{21}), \lambda_{i+n}(D_{23}), 0\} \leq \lambda_{i+n}(W_2 W_2^T) \leq \lambda_i(D_{21}) \leq \lambda_i(\bar{D}_{h_1}),$$

where the first two inequalities uses (F.18) and the fact that $\lambda_{i+n}(W_2 W_2^T) = \lambda_{i+n}(W_2^T W_2)$. Also the last inequality is from the fact that $\lambda_i(\bar{D}_{h_1}) = \max\{\lambda_i(D_{21}), \lambda_i(D_{23}), 0\}, \forall i \in [h_1]$.

For $h_1 \geq i > h_1 - n$, we still have

$$-\infty = \lambda_{i+n}(\bar{D}_{h_1}) \leq \lambda_i(D_{21}) \leq \lambda_i(\bar{D}_{h_1}),$$

Overall, we have

$$\lambda_{i+n}(\bar{D}_{h_1}) \leq \lambda_i(D_{21}) \leq \lambda_i(\bar{D}_{h_1}), \forall i,$$

which is exactly $\bar{D}_{h_1} \succeq_n D_{21}$.

Similarly, for $0 < i \leq h_2 - m$,

$$\lambda_{i+m}(\bar{D}_{h_2}) = \max\{\lambda_{i+m}(D_{21}), \lambda_{i+m}(D_{23}), 0\} \leq \lambda_{i+m}(W_2^T W_2) \leq \lambda_i(D_{23}) \leq \lambda_i(\bar{D}_{h_2}),$$

where the first two inequalities uses (F.18) and the fact that $\lambda_{i+m}(W_2 W_2^T) = \lambda_{i+m}(W_2^T W_2)$. Also the last inequality is from the fact that $\lambda_i(\bar{D}_{h_2}) = \max\{\lambda_i(D_{21}), \lambda_i(D_{23}), 0\}, \forall i \in [h_2]$.

For $h_2 \geq i > h_2 - m$, we still have

$$-\infty = \lambda_{i+m}(\bar{D}_{h_2}) \leq \lambda_i(D_{23}) \leq \lambda_i(\bar{D}_{h_2}),$$

Overall, we have

$$\lambda_{i+m}(\bar{D}_{h_2}) \leq \lambda_i(D_{23}) \leq \lambda_i(\bar{D}_{h_2}), \forall i,$$

which is exactly $\bar{D}_{h_2} \succeq_m D_{23}$. □

*Proof of Lemma F.6.* Notice that $\text{rank}(Z^T Z) \leq r$, hence we consider the eigendecomposition

$$Z^T Z = \sum_{i=1}^{r} \lambda_i(Z^T Z) v_i v_i^T,$$

where $v_i$ are unit eigenvectors of $Z^T Z$. Then we can write

$$A - \lambda_r(Z^T Z) v_i v_i^T - \sum_{i=1}^{r-1} \lambda_i(Z^T Z) v_i v_i^T = B$$

We let $D = A - \lambda_r(Z^T Z) v_i v_i^T$, then by Lemma F.3, we know $A \succeq_1 D$, and $D \succeq_{r-1} B$, which suggests that $\forall i$,

$$\lambda_{i+1}(A) \leq \lambda_i(D) \leq \lambda_i(A) \tag{F.19}$$
$$\lambda_{i+r-1}(D) \leq \lambda_i(B) \leq \lambda_i(D). \tag{F.20}$$

In particular, we have $\lambda_i(D) \leq \lambda_i(A)$ from (F.19) and $\lambda_i(D) \leq \lambda_{i+1-r}(B)$ from (F.20), which suggests

$$\lambda_i(D) \leq \min\{\lambda_i(A), \lambda_{i+1-r}(B)\} = \lambda_i(A \wedge_r B), \forall i.$$

Hence

$$\text{tr}(A \wedge_r B) \geq \text{tr}(D) = \text{tr}(A) - \lambda_r(Z^T Z)\text{tr}(v_i v_i^T) = \text{tr}(A) - \lambda_r(Z^T Z).$$

This proves the first inequality.

For the second the inequality, let $x$ be the unit eigenvector associated with $\lambda_r(ZAZ^T)$, then $\lambda_r(ZAZ^T) = x^T ZAZ^T x$. Now write

$$A - Zxx^T Z^T - Z(I - xx^T)Z^T = B.$$

Let $\tilde{D} = A - Zxx^T Z^T$, then again by Lemma F.3 we have $A \succeq_1 \tilde{D}$, and $\tilde{D} \succeq_{r-1} B$.

Notice that

$$
\begin{aligned}
\tilde{D}^2 &= (A - Zxx^T Z^T)^2 \\
&= A^2 + (Zxx^T Z^T)^2 - AZxx^T Z^T - Zxx^T Z^T A.
\end{aligned}
$$

Taking trace on both side of this equation and using the cyclic property of trace operation lead to

$$\text{tr}(\tilde{D}^2) = \text{tr}\left(A^2\right) + \|Zx\|^4 - 2\lambda_r(ZAZ^T). \tag{F.21}$$

We only need to lower bound $\|Zx\|^4 - \text{tr}(\tilde{D}^2)$, for which we write the eigendecomposition $\tilde{D}$ using eigenpairs $\{(\lambda_i(\tilde{D}), u_i)\}_{i=1}^n$ as

$$\tilde{D} = \sum_{i=1}^n \lambda_i(\tilde{D}) u_i u_i^T = \sum_{j=1}^{n-1} \lambda_i(\tilde{D}) u_i u_i^T + \lambda_n(\tilde{D}) u_n u_n^T.$$

Then we have

$$
\begin{aligned}
\|Zx\|^2 = \text{tr}(Zxx^T Z^T) &= \text{tr}(A) - \text{tr}(\tilde{D}) \\
&= \text{tr}(A) - \sum_{j=1}^{n-1} \lambda_j(\tilde{D}) - \lambda_n(\tilde{D}) \\
&\geq \text{tr}(A) - \sum_{j=1}^{n-1} \lambda_j(A \wedge_r B) - \lambda_n(\tilde{D}) \\
&= \text{tr}(A) - \text{tr}(A \wedge_r B) + \lambda_n(A \wedge_r B) - \lambda_n(\tilde{D}),
\end{aligned}
$$

where the inequality follows similar argument in the previous part of the proof and uses

$$\lambda_i(\tilde{D}) \leq \min\{\lambda_i(A), \lambda_{i+1-r}(B)\} = \lambda_i(A \wedge_r B), \tag{F.22}$$

from the fact that $A \succeq_1 \tilde{D}$, and $\tilde{D} \succeq_{r-1} B$.

Now examine the right-hand side carefully: The first component $\text{tr}(A) - \text{tr}(A \wedge_r B)$ is non-negative because $\lambda_i(A) \geq \lambda_i(A \wedge_r B), \forall i$. The second component $\lambda_n(A \wedge_r B) - \lambda_n(\tilde{D})$ is non-negative as well by (F.22). Therefore the right-hand side is non-negative and we can take square on both sides of the inequality, namely,

$$\|W_1 x\|^4 \geq \left(\text{tr}(A) - \text{tr}(A \wedge_r B) + \lambda_n(A \wedge_r B) - \lambda_n(\tilde{D})\right)^2. \tag{F.23}$$

We also have

$$
\begin{aligned}
\text{tr}(\tilde{D}^2) &= \sum_{i=1}^{n-1} \lambda_i^2(\tilde{D}) + \lambda_n^2(\tilde{D}) \\
&\leq \sum_{i=1}^{n-1} \lambda_i^2(A \wedge_r B) + \lambda_n^2(\tilde{D}) \\
&= \text{tr}\left((A \wedge_r B)^2\right) - \lambda_n^2(A \wedge_r B) + \lambda_n^2(\tilde{D}), \tag{F.24}
\end{aligned}
$$

The inequality holds because for $i = 1, \cdots, n-1$,

$$0 \leq \lambda_{i+1}(A) \leq \lambda_i(\tilde{D}) \leq \lambda_i(A \wedge_r B),$$

where the inequality on the left is from $A \succeq_1 \tilde{D}$ and the inequality on the right is due to (F.22).

With those two inequalities (F.23)(F.24), we have (For simplicity, denote $\lambda_\wedge := \lambda_n(A \wedge_r B), \tilde{\lambda} := \lambda_n(\tilde{D})$)

$$\|W_1 x\|^4 - \operatorname{tr}(\tilde{D}^2) - \left[(\operatorname{tr}(A) - \operatorname{tr}(A \wedge_r B))^2 - \operatorname{tr}\left((A \wedge_r B)^2\right)\right]$$
$$\geq \lambda_\wedge^2 + \tilde{\lambda}^2 - 2\lambda_\wedge \tilde{\lambda} + 2(\lambda_\wedge - \tilde{\lambda})(\operatorname{tr}(A) - \operatorname{tr}(A \wedge_r B)) + \lambda_\wedge^2 - \tilde{\lambda}^2$$
$$= 2\lambda_\wedge^2 - 2\lambda_\wedge \tilde{\lambda} + 2(\lambda_\wedge - \tilde{\lambda})(\operatorname{tr}(A) - \operatorname{tr}(A \wedge_r B))$$
$$= 2(\lambda_\wedge - \tilde{\lambda})(\operatorname{tr}(A) - \operatorname{tr}(A \wedge_r B) + \lambda_\wedge) \geq 0 \,,$$

where the last inequality is due to the facts that $\lambda_\wedge \geq \tilde{\lambda}$ by (F.22) and

$$\operatorname{tr}(A) - \operatorname{tr}(A \wedge_r B) + \lambda_\wedge$$
$$= \sum_{i=1}^{n-1} (\lambda_i(A) - \lambda_i(A \wedge_r B)) + \lambda_n(A) \geq 0 \,.$$

This shows
$$\|Z x\|^4 - \operatorname{tr}(\tilde{D}^2) \geq (\operatorname{tr}(A) - \operatorname{tr}(A \wedge_r B))^2 - \operatorname{tr}\left((A \wedge_r B)^2\right) \,.$$
Finally from (F.21) we have

$$2\lambda_r(Z A Z^T) = \operatorname{tr}\left((A)^2\right) + \|Z x\|^4 - \operatorname{tr}(\tilde{D}^2)$$
$$\geq \operatorname{tr}\left((A)^2\right) - \operatorname{tr}\left((A \wedge_r B)^2\right) + (\operatorname{tr}(A) - \operatorname{tr}(A \wedge_r B))^2 \,.$$

$\square$

To proof Lemma F.7, we need one final lemma

**Lemma F.8.** *Given two real symmetric matrices $A, B$ of order $n$, for any $r \leq n$, if $A \succeq_r B$, then $A \succeq_1 (A \wedge_r B)$ and $(A \wedge_r B) \succeq_{r-1} B$.*

*Proof.* Denote $D := A \wedge_r B$, we show $A \succeq_1 D$ and $D \succeq_{r-1} B$. The following statements holds for any index $i \in [n]$.

First of all, we have
$$\lambda_i(D) = \min\{\lambda_i(A), \lambda_{i+1-r}(B)\} \leq \lambda_i(A) \,, \tag{F.25}$$
and
$$\lambda_{i+1}(A) \leq \min\{\lambda_i(A), \lambda_{i+1-r}(B)\} = \lambda_i(D) \,, \tag{F.26}$$
where $\lambda_{i+1}(A) \leq \lambda_{i+1-r}(B)$ is from $A \succeq_r B$. (F.25)(F.26) together show $A \succeq_1 D$.

Next, notice that
$$\lambda_i(B) \leq \min\{\lambda_i(A), \lambda_{i+1-r}(B)\} = \lambda_i(D) \,, \tag{F.27}$$
where $\lambda_i(B) \leq \lambda_i(A)$ is from $A \succeq_r B$, and
$$\lambda_{i+r-1}(D) = \min\{\lambda_{i+r-1}(A), \lambda_i(B)\} \leq \lambda_i(B) \tag{F.28}$$
(F.27)(F.28) together show $D \succeq_{r-1} B$. $\square$

Then we are ready to prove Lemma F.7

*Proof of Lemma F.7.* Denote $D := A \wedge_r B$. We have shown in Lemma F.8 that $A \succeq_1 D$ and $D \succeq_{r-1} B$.

With the two interlacing relations, we know there exist $x \in \mathbb{R}^{n \times 1}, X \in \mathbb{R}^{n \times (r-1)}$ and some orthogonal matrices $V_1, V_2 \in \mathcal{O}(n)$ such that

$$A - x x^T = V_1 D V_1^T, \qquad D - X X^T = V_2 B V_2^T \,, \tag{F.29}$$

then let $V := V_1 V_2$, we have

$$A - x x^T - V_1 X X^T V_1^T = V_1 V_2 B V_2^T V_1^T = V B V^T \,. \tag{F.30}$$

Notice that

$$xx^T + V_1 X X^T V_1^T = [x \quad V_1 X] \begin{bmatrix} x^T \\ X^T V_1^T \end{bmatrix},$$

then with $Z^T := [x \quad V_1 X] \in \mathbb{R}^{n \times r}$, we can write

$$A - Z^T Z = V_1 V_2 B V_2^T V_1^T = V B V^T .$$

It remains to show $\lambda_r(ZZ^T)$ and $2\lambda_r(ZAZ^T)$ have the exact expressions as stated.

Notice that $A - xx^T = V_1 D V_1^T$, then we have

$$\|x\|^2 = \operatorname{tr}(xx^T) = \operatorname{tr}(A - V_1 D V_1^T) = \operatorname{tr}(A) - \operatorname{tr}(D) . \tag{F.31}$$

Moreover, taking trace on both sides of $(A - xx^T)^2 = (V_1 D V_1^T)^2$ yields

$$\operatorname{tr}\left((A)^2\right) - 2x^T A x + \|x\|^4 = \operatorname{tr}(D^2) ,$$

from which we have

$$2x^T A x = \operatorname{tr}(A) - \operatorname{tr}(D^2) + \|x\|^4 = \operatorname{tr}(A) - \operatorname{tr}(D^2) + (\operatorname{tr}(A) - \operatorname{tr}(D))^2 . \tag{F.32}$$

Finally, notice that the first diagonal entry of

$$ZZ^T = \begin{bmatrix} x^T \\ X^T V_1^T \end{bmatrix} [x \quad V_1 X] = \begin{bmatrix} \|x\|^2 & x^T X \\ X^T x & X^T X \end{bmatrix}$$

is $\|x\|^2$, we have, by (Horn & Johnson, 2012, Corollary 4.3.34),

$$\lambda_r(ZZ^T) \le \|x\|^2 = \operatorname{tr}(A) - \operatorname{tr}(D) = \operatorname{tr}(A) - \operatorname{tr}(A \wedge_r B) .$$

Since we have already shown in Lemma F.6 that

$$\lambda_r(ZZ^T) \ge \operatorname{tr}(A) - \operatorname{tr}(A \wedge_r B) ,$$

we must have the exact equality $\lambda_r(ZZ^T) = \operatorname{tr}(A) - \operatorname{tr}(A \wedge_r B)$.

Similarly, the first diagonal entry of

$$ZAZ^T = \begin{bmatrix} x^T \\ X^T V_1^T \end{bmatrix} A [x \quad V_1 X] = \begin{bmatrix} x^T A x & x^T A X \\ X^T A x & X^T A X \end{bmatrix}$$

is $x^T A x$, then we have, by (Horn & Johnson, 2012, Corollary 4.3.34),

$$2\lambda_r(ZAZ^T) \le 2x^T A x = \operatorname{tr}\left(A^2\right) - \operatorname{tr}\left((A \wedge_r B)^2\right) + (\operatorname{tr}(A) - \operatorname{tr}(A \wedge_r B))^2 .$$

Again, Lemma F.6 shows the inequality in the opposite direction, hence one must take the equality

$$2\lambda_r(ZAZ^T) = x^T A x = \operatorname{tr}\left(A^2\right) - \operatorname{tr}\left((A \wedge_r B)^2\right) + (\operatorname{tr}(A) - \operatorname{tr}(A \wedge_r B))^2 .$$

$\square$

## G    SIMPLIFICATION OF THE BOUND IN THEOREM 2 UNDER UNIMODALITY ASSUMPTION

Consider weights $\{W_1, W_2, W_3\}$ with unimodality index $l^*$, there are three cases:

$\underline{l^* = 1: D_{21} \succeq 0, D_{23} \preceq 0}$

Definiteness of imbalance matrix put rank constraints on the weight matrices:

Since $W_2^T W_2 - W_3 W_3^T = D_{23} \preceq 0$, $\text{rank}(W_3 W_3^T) \leq m$ implies $\text{rank}(D_{23}) \leq m$. ($D_{23}$ can only have negative, if non-zero, eigenvalues and any negative eigenvalue is contributed from $W_3 W_3^T$.)

$\text{rank}(D_{23}) \leq m$ and $D_{23} \preceq 0$ together implies $\text{rank}(W_2^T W_2) \leq m$ ($W_2^T W_2$ having positive invariant subspace with dimension larger than $m$ will give positive eigenvalue to $D_{23}$), which is equivalent to $\text{rank}(W_2^T W_2) \leq m$.

$\text{rank}(W_2^T W_2) \leq m$ forces $\text{rank}(D_{21}) \leq m$. ($D_{22}$ can only have positive, if non-zero, eigenvalues and any positive eigenvalue is contributed from $W_2^T W_2$.)

In summary, we have $\text{rank}(D_{23}) \leq m$ and $\text{rank}(D_{21}) \leq m$, which implies,

$$\lambda_i(D_{23}) = \begin{cases} = 0, & 1 \leq i < h_2 - m + 1 \\ \leq 0, & h_2 - m + 1 \leq i \leq h_2 \end{cases}, \quad \lambda_i(D_{21}) = \begin{cases} \geq 0, & 1 \leq i < m \\ = 0, & m + 1 \leq i \leq h_1 \end{cases}.$$

We also have

$$\bar{D}_{h_1} = \text{diag}\{\max\{\lambda_i(D_{21}), 0\}\}_{i=1}^{h_1} = \text{diag}\{\lambda_i(D_{21})\}_{i=1}^{h_1}, \quad \bar{D}_{h_2} = \text{diag}\{\max\{\lambda_i(D_{21}), 0\}\}_{i=1}^{h_2},$$

Then

$$\bar{D}_{h_1} \wedge_n D_{21} = \bar{D}_{h_1}, \bar{D}_{h_2} \wedge_m D_{23} = \begin{cases} \lambda_i(D_{21}), & 1 \leq i \leq m - 1 \\ 0, & m \leq i < h_2 \\ \lambda_{h_2+1-m}(D_{23}), & i = h_2 \end{cases}.$$

hence $\Delta_{21} = \Delta_{21}^{(2)} = 0$, and

$$\Delta_{23} = \lambda_m(D_{21}) - \lambda_{h_2+1-m}(D_{23})$$
$$\Delta_{23}^{(2)} = \lambda_m^2(D_{21}) - \lambda_{h_2+1-m}^2(D_{23})$$
$$\Delta_{23}^2 + \Delta_{23}^{(2)} = 2\lambda_m(D_{21})(\lambda_m(D_{21}) - \lambda_{h_2+1-m}(D_{23})).$$

$\underline{l^* = 3: D_{23} \succeq 0, D_{21} \preceq 0}$

Similar to previous cases, (by considering unimodal weights $\{W_3^T, W_2^T, W_1^T\}$)

$$\Delta_{23} = \Delta_{23}^{(2)} = 0, \Delta_{21}^2 + \Delta_{21}^{(2)} = 2\lambda_r(D_{23})(\lambda_n(D_{23}) - \lambda_{h_1+1-n}(D_{21})).$$

$\underline{l^* = 2: D_{23} \preceq 0, D_{21} \preceq 0}$

$D_{23}, D_{21}$ being negative semi-definite implies $\text{rank}(D_{21}) \leq n$, $\text{rank}(D_{23}) \leq m$.

In this cases,

$$\bar{D}_{h_1} = 0, \bar{D}_{h_2} = 0,$$

and

$$\bar{D}_{h_1} \wedge_n D_{21} = \begin{cases} 0, & 1 \leq i < h_1 \\ \lambda_{h_1+1-n}(D_{21}), & i = h_1 \end{cases}, \bar{D}_{h_2} \wedge_m D_{23} = \begin{cases} 0, & 1 \leq i < h_2 \\ \lambda_{h_2+1-m}(D_{23}), & i = h_2 \end{cases},$$

then

$$\Delta_{21} = -\lambda_{h_1+1-n}(D_{21}), \qquad\qquad \Delta_{23} = -\lambda_{h_2+1-m}(D_{23}),$$
$$\Delta_{21}^{(2)} = -\lambda_{h_1+1-n}^2(D_{21}), \qquad\qquad \Delta_{23}^{(2)} = -\lambda_{h_2+1-m}^2(D_{23}) = 0.$$

Therefore

$$2\Delta_{21}\Delta_{23} = 2\left(-\lambda_{h_1+1-n}(D_{21})\right)\left(-\lambda_{h_2+1-m}(D_{23})\right), \Delta_{21}^2 + \Delta_{21}^{(2)} = \Delta_{23}^2 + \Delta_{23}^{(2)}.$$

# H    PROOFS FOR DEEP MODELS

We prove Theorem 3 in two parts: First, we prove the lower bound under the unimodality assumption in Section H.1. Then we show the bound for the weights with homogeneous imbalance in Section H.2.

## H.1    LOWER BOUND ON $\lambda_{\min}(\mathcal{T}_{\{W_l\}_{l=1}^L})$ UNDER UNIMODALITY

We need the following two Lemmas (proofs in Section H.3):

**Lemma 4.** *Given $A \in \mathbb{R}^{n \times h}, B \in \mathbb{R}^{h \times m}$, and $D = A^T A - B B^T \in \mathbb{R}^{h \times h}$. If $\mathrm{rank}(A) \leq r$ and $D \succeq 0$, then*

  *1. $\mathrm{rank}(B) \leq r$, and $\mathrm{rank}(D) \leq r$.*

  *2. There exists $Q \in \mathbb{R}^{h \times r}$ with $Q^T Q = I_r$, such that*
$$AQQ^T B = AB, \ AQQ^T A^T = AA^T, \ B^T QQ^T B = B^T B,$$
  *and $\lambda_i(Q^T D Q) = \lambda_i(D)$, $i = 1, \cdots, r$.*

**Lemma 5.** *For $W_1 \in \mathbb{R}^{n \times h_1}, W_2 \in \mathbb{R}^{h_1 \times h_2} \cdots, W_{L-1} \in \mathbb{R}^{h_{L-2} \times h_{L-1}}$ and $W_L \in \mathbb{R}^{h_{L-1} \times h_L}$ such that*
$$D_l = W_l^T W_l - W_{l+1} W_{l+1}^T \succeq 0, \quad l = 1, \cdots, L-1$$
*we have*
$$\lambda_n(W_1 W_2 \cdots W_{L-1} W_{L-1}^T \cdots W_2^T W_1^T) \geq \prod_{i=1}^{L-1} \sum_{l=i}^{L-1} \lambda_n(D_l).$$

Then we can prove the following:

**Theorem H.1.** *For weights $\{W_l\}_{l=1}^L$ with unimodality index $l^*$, we have*
$$\lambda_{\min}\left(\mathcal{T}_{\{W_l\}_{l=1}^L}\right) \geq \prod_{l=1}^{L-1} \tilde{d}_{(i)}. \tag{H.33}$$

*Proof.* Recall that

$$\mathcal{T}_{\{W_l\}_{l=1}^L} E = \sum_{l=1}^L \left(\prod_{i=1}^{l-1} W_i\right) \left(\prod_{i=1}^{l-1} W_i\right)^T E \left(\prod_{i=l+1}^{L+1} W_i\right)^T \left(\prod_{i=l+1}^{L+1} W_i\right), W_0 = I_n, W_{L+1} = I_m.$$

For simplicity, define p.s.d. operators

$$\mathcal{T}_l E := \left(\prod_{i=1}^{l-1} W_i\right) \left(\prod_{i=1}^{l-1} W_i\right)^T E \left(\prod_{i=l+1}^{L+1} W_i\right)^T \left(\prod_{i=l+1}^{L+1} W_i\right), \ l = 1, \cdots, L$$

Then $\mathcal{T}_{\{W_l\}_{l=1}^L} = \sum_{l=1}^L \mathcal{T}_l$.

When $l^* = L$, we have, by Lemma 5,

$$\lambda_{\min}(\mathcal{T}_{\{W_l\}_{l=1}^L}) \geq \lambda_{\min}(\mathcal{T}_L) = \lambda_n(W_1 \cdots W_{L-1} W_{L-1}^T \cdots W_1^T) \geq \prod_{i=1}^{L-1} \sum_{l=i}^{L-1} \lambda_n(D_l) = \prod_{l=1}^{L-1} \tilde{d}_{(i)}.$$

When $l^* = 1$, we have, again by Lemma 5,

$$\lambda_{\min}(\mathcal{T}_{\{W_l\}_{l=1}^L}) \geq \lambda_{\min}(\mathcal{T}_1) = \lambda_m(W_L^T \cdots W_2^T W_2 \cdots W_L) \geq \prod_{i=1}^{L-1} \sum_{l=i}^{L-1} \lambda_m(-D_{L-l})$$
$$= \prod_{i=1}^{L-1} \sum_{l=1}^{L-i} \lambda_m(-D_l)$$
$$= \prod_{i=1}^{L-1} \sum_{l=1}^{i} \lambda_m(-D_l) = \prod_{l=1}^{L-1} \tilde{d}_{(i)}.$$

(To see Lemma 5 applies to the case $l^* = 1$, consider the following

$$W_L^T \to W_1, \cdots, W_{L-l+1}^T \to W_l, \cdots, W_1^T \to W_L,$$

and this naturally leads to $-D_{L-l} \to D_l$. The expressions on the right-hand side of the arrow are those appearing in Lemma 5.)

Now for unimodality index $1 < l^* < L$, we have

$$\lambda_{\min}(\mathcal{T}_{\{W_l\}_{l=1}^L}) \geq \lambda_{\min}(\mathcal{T}_{l^*}) = \lambda_n(W_1 \cdots W_{l^*-1} W_{l^*-1}^T \cdots W_1)\lambda_m(W_L^T \cdots W_{l^*+1}^T W_{l^*+1} \cdots W_L).$$

Now apply Lemma 5 to both $\{W_1, \cdots, W_{l^*-1}, W_{l^*}\}$ and $\{W_L^T, \cdots, W_{l^*+1}^T, W_{l^*}^T\}$, we have

$$\lambda_n(W_1 \cdots W_{l^*-1} W_{l^*-1}^T \cdots W_1) \geq \prod_{i=1}^{l^*-1} \sum_{l=i}^{l^*-1} \lambda_n(D_l) = \prod_{i=1}^{l^*-1} \tilde{d}_{(i)}, \qquad (\text{H.34})$$

and

$$\begin{aligned}
\lambda_m(W_L^T \cdots W_{l^*+1}^T W_{l^*+1} \cdots W_L) &\geq \prod_{i=1}^{L-l^*} \sum_{l=i}^{L-l^*} \lambda_m(-D_{L-l}) \\
&= \prod_{i=1}^{L-l^*} \sum_{l=l^*}^{L-i} \lambda_m(-D_l) \\
&= \prod_{i=l^*}^{L-1} \sum_{l=l^*}^{i} \lambda_m(-D_l) = \prod_{i=l^*}^{L-1} \tilde{d}_{(i)}. \qquad (\text{H.35})
\end{aligned}$$

Combining (H.34) and (H.35), we have

$$\lambda_n(W_1 \cdots W_{l^*-1} W_{l^*-1}^T \cdots W_1)\lambda_m(W_L^T \cdots W_{l^*+1}^T W_{l^*+1} \cdots W_L) \geq \prod_{i=1}^{L-1} \tilde{d}_{(i)}, \qquad (\text{H.36})$$

which leads to $\lambda_{\min}(\mathcal{T}_{\{W_l\}_{l=1}^L}) \geq \prod_{i=1}^{L-1} \tilde{d}_{(i)}$. The proof is complete as we have shown $\lambda_{\min}(\mathcal{T}_{\{W_l\}_{l=1}^L}) \geq \prod_{i=1}^{L-1} \tilde{d}_{(i)}$ for any unimodality index $l^* \in [L]$. $\qquad \square$

## H.2 Lower bound on $\lambda_{\min}(\mathcal{T}_{\{W_l\}_{l=1}^L})$ under homogeneous imbalance

We need the following Lemma (proof in Section H.3):

**Lemma H.2.** *Given any set of scalars $\{w_l\}_{l=1}^L$ such that $d_{(i)} := w_i^2 - w_L^2 \geq 0, i = 1, \cdots, L-1$, we have*

$$\sum_{l=1}^L \prod_{i \neq l} w_i^2 = \sum_{l=1}^L \frac{w^2}{w_l^2} \geq \sqrt{\left(\prod_{i=1}^{L-1} d_{(i)}\right)^2 + \left(Lw^{2-2/L}\right)^2}, \qquad (\text{H.37})$$

*where $w = \prod_{l=1}^L w_l$.*

Then we can prove the following:

**Theorem H.3.** *For weights $\{W_l\}_{l=1}^L$ with homogeneous imbalance, we have*

$$\lambda_{\min}\left(\mathcal{T}_{\{W_l\}_{l=1}^L}\right) \geq \sqrt{\left(\prod_{l=1}^{L-1} \tilde{d}_{(i)}\right)^2 + \left(L\sigma_{\min}^{2-2/L}(W)\right)^2}, \quad W = \prod_{l=1}^L W_l. \qquad (\text{H.38})$$

*Proof.* When all imbalance matrices are zero matrices, this is the balanced case (Arora et al., 2018b) and $\lambda_{\min}\left(\mathcal{T}_{\{W_l\}_{l=1}^L}\right) = L\sigma_{\min}^{2-2/L}(W)$. Here we only prove the case when some $d_l \neq 0$.

Notice that given the homogeneous imbalance constraint

$$W_l^T W_l - W_{l+1} W_{l+1}^T = d_l I,$$

$W_l^T W_l$ and $W_{l+1} W_{l+1}^T$ must be co-diagonalizable: If we have $Q^T Q = I$ such that $Q^T W_l^T W_l Q$ is diagonal, then $Q^T W_{l+1} W_{l+1}^T Q$ must be diagonal as well since $Q^T W_l^T W_l Q - Q^T W_{l+1} W_{l+1}^T Q = d_l I$.

Moreover, if the diagonal entries of $Q^T W_l^T W_l Q$ are in decreasing order, then so are those of $Q^T W_{l+1} W_{l+1}^T Q$ because the latter is the shifted version of the former by $d_l$.

This suggests that all $W_l, l = 1, \cdots, L$ have the same rank and one has the following decomposition of the weights:

$$W_l = Q_{l-1} \Sigma_l Q_l^T \,, \tag{H.39}$$

Here, $\Sigma_l, l = 1, \cdots, L$ are diagonal matrix of size $k = \min\{n, m\}$ whose entries are in decreasing order. And $Q_l \in \mathbb{R}^{h_l \times \min\{n,m\}}$ with $Q_l^T Q_l = I$. ($h_0 = n, h_L = m$). From such decomposition, we have

$$W = W_1 \cdots W_L = Q_0 \Sigma_1 Q_1^T Q_1 \Sigma_2 Q_2^T \cdots Q_{L-1} \Sigma_L Q_L^T = Q_0 \left( \prod_{l=1}^L \Sigma_l \right) Q_L^T \,, \tag{H.40}$$

thus

$$\sigma_{\min}(W) = \prod_{l=1}^L \lambda_{\min}(\Sigma_l) \,. \tag{H.41}$$

Regarding the imbalance, we have

$$Q_l^T (W_l^T W_l - W_{l+1} W_{l+1}^T) Q_l = d_l I \quad \Rightarrow \quad \Sigma_l^2 - \Sigma_{l+1}^2 = d_l I \,, \tag{H.42}$$

which suggests that

$$\lambda_{\min}^2(\Sigma_l) - \lambda_{\min}^2(\Sigma_{l+1}) = d_l, l = 1, \cdots, L - 1 \,. \tag{H.43}$$

Now consider the set of scalars $\{w_l\}_{l=1}^L$:

$$w_l = \lambda_{\min}(\Sigma_l), l = 1, \cdots, l^* - 1$$
$$w_l = \lambda_{\min}(\Sigma_{l+1}), l = l^*, \cdots, L - 1$$
$$w_L = \lambda_{\min}(\Sigma_{l^*}) \,.$$

Then $\{w_l\}_{l=1}^L$ satisfy the assumption in Lemma H.2:

$$w_i^2 - w_L^2 = \tilde{d}_{(i)} \geq 0, i = 1, \cdots, L - 1 \,, \tag{H.44}$$

where $\tilde{d}_{(i)}$ is precisely the cumulative imbalance. Then Lemma H.2 gives ((H.41) is also used here)

$$\sum_{l=1}^L \prod_{i \neq l} w_i^2 \geq \sqrt{\left( \prod_{i=1}^{L-1} \tilde{d}_{(i)} \right)^2 + \left( L \sigma_{\min}^{2-2/L}(W) \right)^2} \tag{H.45}$$

Recall that

$$\mathcal{T}_{\{W_l\}_{l=1}^L} E = \sum_{l=1}^L \left( \prod_{i=0}^{l-1} W_i \right) \left( \prod_{i=0}^{l-1} W_i \right)^T E \left( \prod_{i=l+1}^{L+1} W_i \right)^T \left( \prod_{i=l+1}^{L+1} W_i \right) \,, W_0 = I_n, W_{L+1} = I_m \,.$$

For simplicity, define p.s.d. operators

$$\mathcal{T}_l E := \left( \prod_{i=0}^{l-1} W_i \right) \left( \prod_{i=0}^{l-1} W_i \right)^T E \left( \prod_{i=l+1}^{L+1} W_i \right)^T \left( \prod_{i=l+1}^{L+1} W_i \right) \,, l = 1, \cdots, L$$

Then $\mathcal{T}_{\{W_l\}_{l=1}^L} = \sum_{l=1}^L \mathcal{T}_l$.

Notice that the summand $\prod_{i \neq l} w_i^2$ exactly corresponds to one of $\lambda_{\min}(\mathcal{T}_l)$. For example,

$$\lambda_{\min}(\mathcal{T}_1) = \lambda_{\min}(W_L^T \cdots W_2^T W_2 \cdots W_L) = \lambda_{\min} \left( Q_L^T \left( \prod_{l=2}^L \Sigma_l^2 \right) Q_L \right) = \prod_{i \neq 1} w_i^2 \,. \tag{H.46}$$

More precisely, we have

$$\lambda_{\min}(\mathcal{T}_l) = \prod_{i \neq l} w_i^2, \quad l < l^*$$

$$\lambda_{\min}(\mathcal{T}_l) = \prod_{i \neq l-1} w_i^2, \quad l > l^*$$

$$\lambda_{\min}(\mathcal{T}_l) = \prod_{i \neq L} w_i^2, \quad l = l^*.$$

Therefore, we finally have

$$\lambda_{\min}(\mathcal{T}_{\{W_l\}_{l=1}^L}) \geq \sum_{l=1}^L \lambda_{\min}(\mathcal{T}_l) = \sum_{l=1}^L \prod_{i \neq l} w_i^2 \geq \sqrt{\left(\prod_{i=1}^{L-1} \tilde{d}_{(i)}\right)^2 + \left(L\sigma_{\min}^{2-2/L}(W)\right)^2}. \quad \text{(H.47)}$$

$\square$

## H.3 PROOFS FOR AUXILIARY LEMMAS

*Proofs for Lemma 5.* The proof is rather simple when $n = h_1 = h_2 = \cdots = h_{L-1}$: Notice that

$$\lambda_n(W_1 W_2 \cdots W_{L-1} W_{L-1}^T \cdots W_2^T W_1^T)$$
$$\geq \lambda_n(W_{L-1} W_{L-1}^T) \cdot \lambda_n(W_1 W_2 \cdots W_{L-2} W_{L-2}^T \cdots W_2^T W_1^T)$$
$$\geq \lambda_n(W_{L-1} W_{L-1}^T) \cdot \lambda_n(W_{L-2} W_{L-2}^T) \cdot \lambda_n(W_1 W_2 \cdots W_{L-3} W_{L-3}^T \cdots W_2^T W_1^T)$$
$$\cdots$$
$$\geq \prod_{i=1}^{L-1} \lambda_n(W_i W_i^T).$$

Then it remains to show that $\lambda_n(W_i W_i^T) \geq \sum_{l=i}^{L-1} \lambda_n(D_l)$ for $i = 1, \cdots, L-1$.

Suppose $\lambda_n(W_k W_k^T) \geq \sum_{l=k}^{L-1} \lambda_l(D)$ for some $k \in [L-1]$, then we have

$$\begin{aligned}
\lambda_n(W_{k-1} W_{k-1}^T) &= \lambda_n(W_{k-1}^T W_{k-1}) \\
&= \lambda_n(W_k W_k^T + D_{k-1}) \\
&\geq \lambda_n(W_k W_k^T) + \lambda_n(D_{k-1}) \\
&\geq \sum_{l=k}^{L-1} \lambda_n(D_l) + \lambda_n(D_{k-1}) = \sum_{l=k-1}^{L-1} \lambda_n(D_l).
\end{aligned}$$

Therefore, we only need to show $\lambda_n(W_{L-1} W_{L-1}^T) \geq \lambda_n(D_{L-1})$ then the rest follows by the induction above. Indeed

$$\lambda_n(W_{L-1} W_{L-1}^T) = \lambda_n(W_{L-1}^T W_{L-1}) = \lambda_n(W_L W_L^T + D_{L-1}) \geq \lambda_n(D_{L-1}),$$

which finishes the proof for the case of $n = h_1 = h_2 = \cdots = h_{L-1}$.

When the above assumptions does not hold, Lemma 4 allows us to related the set of weights $\{W_l\}_{l=1}^L$ to the one $\{\tilde{W}_l\}_{l=1}^L$ that satisfy the equal dimension assumption. More specifically, apply Lemma 4 using each imbalance constraint

$$D_l = W_l^T W_l - W_{l+1} W_{l+1}^T \succeq 0, \quad l = 1, \cdots, L-1,$$

to obtain a $Q_l \in \mathbb{R}^{h_l \times n}$ that has all the property in Lemma (4). Use these $Q_l, l = 1, \cdots, L-1$ to define

$$\tilde{W}_l = Q_{l-1}^T W_l Q_l, l = 1, \cdots, L,$$
$$\tilde{D}_l = \tilde{W}_l^T \tilde{W}_l - \tilde{W}_{l+1}^T \tilde{W}_{l+1}, l = 1, \cdots, L-1,$$

where $Q_0 = I, Q_L = I$. Now $\{\tilde{W}_l\}_{l=1}^L$ satisfies the assumption that $n = h_1 = \cdots = h_{L-1}$, then

$$\lambda_n(\tilde{W}_1 \tilde{W}_2 \cdots \tilde{W}_{L-1} \tilde{W}_{L-1}^T \cdots \tilde{W}_2^T \tilde{W}_1^T) \geq \prod_{i=1}^{L-1} \sum_{l=i}^{L-1} \lambda_n(\tilde{D}_l). \tag{H.48}$$

Using the properties of $Q_l \in \mathbb{R}^{h_l \times n}, l = 1, \cdots, L-1$, we have

$$\begin{aligned}
&\lambda_n(\tilde{W}_1 \tilde{W}_2 \cdots \tilde{W}_{L-1} \tilde{W}_{L-1}^T \cdots \tilde{W}_2^T \tilde{W}_1^T) \\
&= \lambda_n(W_1 Q_1 Q_1^T W_2 Q_2 \cdots Q_{L-2}^T W_{L-1} Q_{L-1} Q_{L-1}^T W_{L-1}^T Q_{L-2}^T \cdots Q_2^T W_2^T Q_1 Q_1^T W_1^T) \\
&= \lambda_n(W_1 W_2 \cdots W_{L-1} W_{L-1}^T \cdots W_2^T W_1^T),
\end{aligned}$$

and

$$\prod_{i=1}^{L-1} \sum_{l=i}^{L-1} \lambda_n(\tilde{D}_l) = \prod_{i=1}^{L-1} \sum_{l=i}^{L-1} \lambda_n(Q_l^T D_l Q_l) = \prod_{i=1}^{L-1} \sum_{l=i}^{L-1} \lambda_n(D_l).$$

Therefore, (H.48) is exactly

$$\lambda_n(W_1 W_2 \cdots W_{L-1} W_{L-1}^T \cdots W_2^T W_1^T) \geq \prod_{i=1}^{L-1} \sum_{l=i}^{L-1} \lambda_n(D_l). \tag{H.49}$$

$\square$

*Proofs for Lemma 4.* Since $\text{rank}(A) \leq r$, $A$ has a compact SVD $A = P\Sigma_A Q^T$ such that $Q \in \mathbb{R}^{h \times r}$ and $Q^T Q = I_r$.

This is exactly $Q$ we are looking for. Let $Q_\perp Q_\perp^T = I_h - QQ^T$ be the projection onto the subspace orthogonal to the columns of $Q$. Then

$$D = A^T A - BB^T \Rightarrow Q_\perp^T D Q_\perp = Q_\perp^T A^T A Q_\perp - Q_\perp^T BB^T Q_\perp \Rightarrow Q_\perp^T D Q_\perp + Q_\perp^T BB^T Q_\perp = 0.$$

$Q_\perp^T D Q_\perp$ and $Q_\perp^T BB^T Q_\perp$ are two p.s.d. matrices whose sum is zero, which implies

$$Q_\perp^T D Q_\perp = 0, \quad D Q_\perp = 0, \quad Q_\perp^T BB^T Q_\perp = 0, \quad B^T Q_\perp = 0.$$

$Q_\perp^T D Q_\perp = 0$ shows that the nullspace of $D$ has at least dimension $h - r$, i.e., $\text{rank}(D) \leq r$.

Moreover

$$\begin{aligned}
AQQ^T B &= A(I_h - Q_\perp Q_\perp^T) B = AB \\
AQQ^T A^T &= A(I_h - Q_\perp Q_\perp^T) A^T = AA^T \\
B^T QQ^T B &= B^T (I_h - Q_\perp Q_\perp^T) B = B^T B
\end{aligned}$$

The last equality $B^T B = B^T QQ^T B$ shows that $\text{rank}(B) \leq r$.

Lastly, we have, for $i = 1, \cdots, r$,

$$\lambda_i(Q^T D Q) = \lambda_i(QQ^T D) = \lambda_i((I_h - Q_\perp Q_\perp^T)D) = \lambda_i(D).$$

$\square$

Before proving Lemma H.2, we state a Lemma that will be used in the proof.

**Lemma H.4.** *Given positive $x_i, i = 1, \cdots, n$, we have*

$$\sum_{i=1}^n x_i \geq n \left( \prod_{i=1}^n x_i \right)^{1/n}.$$

*Proof.* This is from the fact that arithmetic mean of $\{x_i\}_{i=1}^n$ is greater than the geometric mean of $\{x_i\}_{i=1}^n$. $\square$

We are ready to prove Lemma H.2.

*Proof of Lemma H.2.* We denote

$$\tau_{\{w_l\}_{i=1}^L} := \sum_{l=1}^L \prod_{i \neq l} w_i^2 \tag{H.50}$$

Notice that $w_i^2 = w_L^2 + \sum_{j=i}^{L-1}(w_j^2 - w_{j+1}^2) = w_L^2 + d_{(i)}$. Let $d_{(L)} = 0$, we write the expression for $\tau$ as

$$\tau_{\{w_l\}_{i=1}^L} = \sum_{l=1}^L \prod_{i \neq l} w_i^2 = \sum_{l=1}^L \prod_{i \neq l} \left(w_L^2 + d_{(i)}\right) := \tau(w_L^2; \{d_{(i)}\}_{i=1}^{L-1}).$$

Therefore, when fixing $\{d_{(i)}\}_{i=1}^{L-1}$, $\tau$ can be viewed as a function of $w_L^2$.

**When** $w = 0$: one of $w_l$ must be zero, and because $w_L^2$ has the least value among all the weights, we know $w_L^2 = 0$. Then

$$\tau_{\{w_l\}_{i=1}^L} = \tau(0; \{d_{(i)}\}_{i=1}^{L-1}) = \prod_{i=1}^{L-1} d_{(i)},$$

i.e. we actually have equality when $w = 0$.

**When** $w \neq 0$: then $w^2 \neq 0$ and we write

$$w^2 = \prod_{l=1}^L w_l^2 = w_L^2 \prod_{l=1}^{L-1} \left(w_L^2 + d_{(l)}\right) := p(w_L^2; \{d_{(i)}\}_{i=1}^{L-1}),$$

which shows $w^2$ is a function of $w_L^2$ when $\{d_{(i)}\}_{i=1}^{L-1}$ are fixed. Here we use $p$ to denote $w^2$ for simplicity. Moreover, function $p\colon \mathbb{R}_{\geq 0} \to \mathbb{R}_{\geq 0}$ has differentiable inverse $p^{-1}$ as long as $p > 0$, because

$$\frac{dp}{dw_L^2} = \sum_{l=1}^L \prod_{i \neq l} \left(w_L^2 + d_{(i)}\right) = \sum_{l=1}^L \prod_{i \neq l} w_i^2 \overset{\text{(Lemma H.4)}}{\geq} L \left(p^{L-1}\right)^{1/L} > 0,$$

and inverse function theorem (Rudin, 1953) shows the existence of differentiable inverse. Whenever, $p^{-1}$ exists, it derivative is

$$\frac{dw_L^2}{dp} = \left(\sum_{l=1}^L \prod_{i \neq l} \left(w_L^2 + d_{(i)}\right)\right)^{-1} = \tau^{-1}.$$

Now pick any $0 < p_0 \leq w^2$ we have, by Fundamental Theorem of Calculus,

$$\begin{aligned} \tau_{\{w_l\}_{l=1}^L}^2 &= \tau^2(p^{-1}(w^2); \{d_{(i)}\}_{i=1}^{L-1}) \\ &= \tau^2(p^{-1}(p_0); \{d_{(i)}\}_{i=1}^{L-1}) + \int_{p^{-1}(p_0)}^{p^{-1}(w^2)} \frac{d}{dw_L^2} \tau^2(w_L^2; \{d_{(i)}\}_{i=1}^{L-1}) dw_L^2 \end{aligned}$$

For the first part, we have

$$\begin{aligned} &\tau^2(p^{-1}(p_0); \{d_{(i)}\}_{i=1}^{L-1}) \\ &= \left(\sum_{l=1}^L \prod_{i \neq l} \left(p^{-1}(p_0) + d_{(i)}\right)\right)^2 \geq \left(\prod_{i \neq L} \left(p^{-1}(p_0) + d_{(i)}\right)\right)^2 \geq \left(\prod_{i=1}^{L-1} d_{(i)}\right)^2, \end{aligned}$$

and for the second part, we have

$$\int_{p^{-1}(p_0)}^{p^{-1}(w^2)} \frac{d}{dw_L^2} \tau^2 dw_L^2$$

$$= \int_{p^{-1}(p_0)}^{p^{-1}(w^2)} 2\tau \frac{d}{dw_L^2} \tau dw_L^2$$

$$= \int_{p^{-1}(p_0)}^{p^{-1}(w^2)} 2\tau \sum_{l=1}^{L} \sum_{i \neq l} \prod_{j \neq i, j \neq l} (w_L^2 + d_{(j)}) dw_L^2$$

$$= \int_{p^{-1}(p_0)}^{p^{-1}(w^2)} 2\tau \sum_{l=1}^{L} \sum_{i \neq l} \frac{p}{w_i^2 w_l^2} dw_L^2$$

$$(\text{Lemma H.4}) \geq \int_{p^{-1}(p_0)}^{p^{-1}(w^2)} 2\tau L(L-1) \left( \prod_{l=1}^{L} \prod_{i \neq l} \frac{p}{w_i^2 w_l^2} \right)^{\frac{1}{L(L-1)}} dw_L^2$$

$$= \int_{p^{-1}(p_0)}^{p^{-1}(w^2)} 2\tau L(L-1) \left( \frac{p^{L(L-1)}}{p^{2L-2}} \right)^{\frac{1}{L(L-1)}} dw_L^2$$

$$= \int_{p^{-1}(p_0)}^{p^{-1}(w^2)} 2\tau L(L-1) p^{1-2/L} dw_L^2$$

$$(dw_L^2 = \tau^{-1} dp) = \int_{p_0}^{w^2} 2L(L-1) p^{1-2/L} dp = L^2 p^{2-2/L} \Big|_{p_0}^{w^2} = \left( Lw^{2-2/L} \right)^2 - L^2 p_0^{2-2/L}.$$

Overall, for any $0 < p_0 \leq w^2$, we have

$$\tau_{\{w_l\}_{l=1}^{L}}^2 \geq \left( \prod_{i=1}^{L-1} d_{(i)} \right)^2 + \left( Lw^{2-2/L} \right)^2 - L^2 p_0^{2-2/L}.$$

Let $p_0 \to 0$, we have $\tau^2 \geq \left( \prod_{i=1}^{L-1} d_{(i)} \right)^2 + \left( Lw^{2-2/L} \right)^2$, i.e.

$$\tau \geq \sqrt{\left( \prod_{i=1}^{L-1} d_{(i)} \right)^2 + \left( Lw^{2-2/L} \right)^2}.$$

$\square$

