# OpenReview forum: "On the Convergence of Gradient Flow on Multi-layer Linear Models"
_ICLR.cc/2023/Conference — Submitted to ICLR 2023_

### Official Review · Reviewer_QQB2 · 2022-10-21

**Confidence:** 3
**Correctness:** 4
**Technical Novelty And Significance:** 3
**Empirical Novelty And Significance:** Not applicable
**Recommendation:** 6

**Clarity, Quality, Novelty And Reproducibility:**

- The paper is well-written with a sufficient literature review. The main ideas and techniques of the work were developed based on well-founded rationales.
- The mathematical analyses of the work are rigorous and seem correct.
- The paper obtains several new technical results/improvements.

**Strength And Weaknesses:**

### Strengths

- The paper is well-written with a sufficient literature review. The main ideas and techniques of the work were developed based on well-founded rationales.
- The mathematical analyses of the work are rigorous and seem correct. The framework of the paper is general and broadly applicable (covers several existing initialization schemes; general loss functions)
- The paper obtains several new technical results/improvements
    - The derived bounds (Theorems 2,3) characterize the general effect of weight imbalance on convergence. Previous works on this aspect focused on two-layer models or when all imbalance matrices are positive semi-definite.
    - The analysis applies to general loss functions and thus can be used to study classification tasks (Theorem 4). Existing works mostly focused on l2 loss for regression tasks.
    - Three-layer model: Theorem 2 is stronger than previous work (Yun et. al.(2020)) in that it doesn’t force the partial ordering of the weights for convergence. The paper also proves that the bound in this case is optimal.
    - Deep linear models:
        - The bound of Theorem 3 dominates and unifies two existing bounds on deep networks (Arora et. al.(2018a), Yun et. al.(2020)) and characterizes the general effect of weight imbalance on convergence
        - The unimodality condition seems to be novel and contains that of Yun et. al.(2020) as a special case
    - Some technical constructions of the proof of Theorem 2 (interlacing of spectra of two matrices whose difference is positive definite and low-rank; explicit construct of the optimal solution to the lower bound) might be of general interest.

### Weaknesses

- None noted

**Summary Of The Paper:**

The paper aims to analyze the convergence of gradient flow of multi-layer linear models. The paper deposit that in general, the convergence rate depends on two trajectory-specific quantities: the imbalance matrices (which measure the difference between the weights of adjacent layers) and on the least singular values of the weight product. The framework of the paper is designed to cover several existing initialization schemes and applies to both regression and classification.

**Summary Of The Review:**

The paper addresses a meaningful question. The work obtains several new technical results, and provides some useful insights about the effect of weight imbalance on convergence for deep linear models.

Update: The reviewers' discussions come to a general assessments that while the contributions of the manuscripts are significant and somewhat new, most of the technical improvements are marginal with some concerns about the realisticness of some of the assumptions. My final score was adjusted from 8 to 6 to reflect this summary of the discussions.

---

> ### Author Response · Authors · 2022-11-19
> **Response to Reviewer QQB2**
>
> Thank you very much for your positive review and we very much appreciate your support of our work and recognition of our contribution! Please feel free to post more comments and questions during the discussion period. Thank you!

---

### Official Review · Reviewer_5qgv · 2022-10-25

**Confidence:** 4
**Correctness:** 4
**Technical Novelty And Significance:** 3
**Empirical Novelty And Significance:** 3
**Recommendation:** 6

**Clarity, Quality, Novelty And Reproducibility:**

Clarity: High
Quality: High
Novelty: Fair. It seems that the work is an extension of Min et al. 2022 to multi-layer setting.

**Strength And Weaknesses:**

Strength:
1. The paper is well written and clearly motivated.
2. The obtained convergence result is quite general and can be applied to a wide range of imbalanced initializations.

Weakness:
1. Limited to linear networks

**Summary Of The Paper:**

This paper proves the exponential convergence of gradient flow on multi-layer linear models in which the loss function f satisfies the gradient dominance property. It also provides a lower bound on the convergence rate that depends on the imbalance matrices and the least singular value of the weight product.

**Summary Of The Review:**

The paper provides the exponential convergence result for the gradient flow of multi-layer linear networks under a gradient dominance assumption. It applies to a wide range of imbalanced initializations. It is technically correct and the obtained result is new. The novelty of this paper is fair and it builds upon several existing works on overparametrized linear networks. The impact of this work is not clear as it only applies to linear networks.

---

> ### Author Response · Authors · 2022-11-19
> **Response to Reviewer 5qgv**
>
> We thank the reviewer for the positive review.
>
> We understand the reviewer's concern about the significance of our results as we have only studied linear networks. However, it is our view that to gain deep insights into highly non-trivial phenomena one needs to first thoroughly understand simpler cases. The fact is that, today, we do not even have a good understanding of how "simple" multi-layer linear models can be successfully trained. Our paper greatly improves existing results on the deep linear networks, and thus makes a significant contribution. As Reviewer QQB2 pointed out, "_The paper addresses a meaningful question. The work obtains several new technical results, and provides some useful insights..._" ; we certainly agree with this view.
>
> We do not expect that our results can be easily generalized to ReLU networks, but *they can still provide insights on training of nonlinear, deep networks*. Precisely, the imbalance matrices $D_l$ are still defined for ReLU activations, and their diagonals are invariant under the subgradient flow differential inclusion. Such invariance could potentially be used for studying convergence for non-linear networks.
>
> We hope these clarifications are helpful. Please feel free to post more comments and questions during the discussion period. Thank you!

---

### Official Review · Reviewer_h9zp · 2022-11-08

**Confidence:** 4
**Correctness:** 2
**Technical Novelty And Significance:** 3
**Empirical Novelty And Significance:** Not applicable
**Recommendation:** 3

**Clarity, Quality, Novelty And Reproducibility:**

The mathematical writing is clear.  There seem to be a number of substantial new ideas in the paper.

**Strength And Weaknesses:**

The way that the PL condition is used to formulate the results is
unfamiliar to me, and seems nice.

Where the authors write ``cannot explain well the training efficiency
in practice'', I don't find this completely convincing, because people
don't use extremely small initializations in practice.  Of course it
is interesting to determine how the convergence time depends on the
size of the initialization.

It is not clear to me how good the lower bound in Theorem 2 is.  It is
consistent with my current understanding, such as it is, that the
bound of Theorem 2 is often zero, which, when true, makes Corollary 1
vacuous.  It would be helpful to provide some examples of where
Theorem 2 provides a good bound.

I don't think that some of the claims made in the text after Theorem 2
are justified by the result.  For example, since Theorem 2 only has a
lower bound, I don't see that the claim that they ``fully characterize
the effect of imbalance on the convergence of three-layer networks''
is justified.  Also, to show that ``such a partial ordering is not
necessary'', you would need to provide an example of where the RHS of
(17) was positive without such a partial ordering.  Similarly, I don't
see that claims made in the intro about characterization are
justified.

The condition defined in Definition 3 seems like a
strong assumption to me.  For example, it seems like it
is very unlikely to be satisfied, or even nearly satisfied,
by a random initialization.  I think that more justification is
needed that this is an interesting condition to study.
Also, while it is more general than the condition used
in [2], it seems to make available a similar set of tools.
It also reads to me as being crafted to be able to apply
Weyl's inequality, rather than to capture a useful product
of natural, and random, initializations.  Roughly speaking,
it looks to me that unimodality is ``looking under the
lamppost''.

Some more comparison with prior work would be helpful.  For example, I
believe that the results of this paper are incomparable in strength
with the results in [3]-[5], which touch on an overlapping set of
issues.  I think that the claim that `` the convergence analysis for
imbalanced networks not in the kernel regime has only been studied for
specific initializations'' is not correct, though I do feel that,
despite this, the authors' comparison with [2] seems fair overall.

Here are a couple of smaller points.

The unimodality index is not defined until Definition 3, but is
used earlier in the proof of Theorem 2 -- the authors should move
it earlier.

Before (13), the authors indicate that they are going to define
$D_1$ and $D_2$, but then they define $D_{21}$ and $D_{23}$.

I will carefully read and consider the authors' reply.

[1] Hancheng Min, Salma Tarmoun, René Vidal, and Enrique
Mallada. Convergence and implicit bias of gradient flow on
overparametrized linear networks. arXiv preprint arXiv:2105.06351,
2022.

[2] Yun, Chulhee, Shankar Krishnan, and Hossein Mobahi. "A unifying
view on implicit bias in training linear neural networks."
International Conference on Learning Representations. 2020.

[3] Jin, Chi, et al. "How to escape saddle points efficiently."
International Conference on Machine Learning. PMLR, 2017.

[4] Hu, Wei, Lechao Xiao, and Jeffrey Pennington. "Provable Benefit of
Orthogonal Initialization in Optimizing Deep Linear Networks."
International Conference on Learning Representations. 2019.

[5] Zou, Difan, Philip M. Long, and Quanquan Gu. "On the Global
Convergence of Training Deep Linear ResNets." International Conference
on Learning Representations. 2019.

**Summary Of The Paper:**

This paper extends the interesting results of [1] on the convergence
of gradient flow for two-layer linear networks to the case of deep
networks, and to a wider variety of loss functions, obtaining
bounds on the rate of convergence in terms of notions of the
``imbalance'' of the initialization of the network.  They also extend
results from [2] on deep networks to a wider variety of
initializations, again exposing a wider variety of effects.
Their bounds for three-layer networks are quite general, and they
provide bounds for deeper networks whose initializations satisfy
a condition like one used in [2], but more general.

[1] Hancheng Min, Salma Tarmoun, René Vidal, and Enrique
Mallada. Convergence and implicit bias of gradient flow on
overparametrized linear networks. arXiv preprint arXiv:2105.06351,
2022.

[2] Yun, Chulhee, Shankar Krishnan, and Hossein Mobahi. "A unifying
view on implicit bias in training linear neural networks."
International Conference on Learning Representations. 2020.

**Summary Of The Review:**

The strength of the main results was unclear to me, and I felt that some claims made in the paper were not justified by the results.

---

> ### Author Response · Authors · 2022-11-19
> **Response to Reviewer h9zp (1/2)**
>
> We respectfully disagree with the reviewer's comment on the strength of our results (Theorem 2 and Theorem 3). That being said, the reviewer's comments have helped us better articulate their significance. We have modified the paper accordingly to provide a better explanation and interpretation of our theoretical results that highlight their significance and a better comparison with prior work. We thank the reviewer for it.
>
> **Small initialization**: We think our writing in the original draft regarding small initialization might have not been sufficiently clear, which led to a misunderstanding on this issue. We meant to say "The analysis for small initialization, while insightful in understanding the implicit bias of neural network training, is not suitable for understanding the training efficiency in practice since small initialization is rarely implemented due to its slow convergence." We have modified our discussion on small initialization accordingly.
>
> **Bound in Theorem 2**: We would like to point out that the strength of Theorem 2 is that it does not require ANY assumption on the initialization. Much of the existing work assumes some special property of the initialization, and shows exponential convergence". For example, the NTK initialization [4,7] assumes random Gaussian initialization + extremely large hidden layer width, the small initialization [9] assumes vanishing initialization scale, and the balanced initialization [8] assumes zero imbalance). Therefore, the applicability of such results is limited to the prescribed initializations. On the contrary, our result on the exponential convergence for three-layer networks applies to ANY initialization (and almost any choice of network width), provided that our bound is positive. Such assumption-free bound requires an analysis of the learning dynamic that does not exploit properties of a specific trajectory (chosen by proper initialization). To our knowledge, this has only been achieved by [1] for two-layer linear networks. Our bound in Theorem 2 is the first one achieving such a result for three-layer linear networks.
>
> Our bound is useful since we can 1) test if some initialization leads to convergence, and 2) characterize a wide range of initialization for which exponential convergence is guaranteed. Most existing analyses on deep linear networks can not achieve this because they assume specific initialization, a priori. While we understand your concern that our bound in Theorem 2 might be zero, the key point of our analysis is to derive sufficient conditions under which the bound is positive, please refer to the remark "Implication on convergence" (highlighted in blue) after Theorem 2. Simply speaking, the bound is positive if the imbalance matrices $D_{21}$ and $D_{23}$ have a certain number of negative eigenvalues. Notice that the imbalance matrices are defined to be the difference between two positive semidefinite matrices, such a condition is easy to be satisfied. Here we are discovering new initialization that leads to convergence solely by finding conditions for which the bound is positive. Lastly, we note that our remark "Connection to results for three-layer" in Section 4.2 also shows another condition under which the bound is positive.
>
> Regarding our claim "we fully characterize the effect of imbalance on the convergence of three-layer networks", we agree that it is an overstatement and we have modified it to: "The convergence of multi-layer linear networks under balanced initialization ($D_{l}=0,\forall l$) has been studied in Arora et al., and our result is complementary as we study the effect of non-zero imbalance on the convergence of three-layer networks."

---

> > ### Author Response · Authors · 2022-11-19
> > **Response to Reviewer h9zp (2/2)**
> >
> > **Unimodality assumption**: First, we note that our results for three-layer networks do not require a unimodality assumption. For deep networks, as we discussed in "Bound in Theorem 2", most existing work makes unrealistic assumptions such as extremely large hidden layer width (required by [4,7]), small initialization (required by [9]), diagonal structure (required by [10]), spectral initialization (required by [11]), balanced weights (required by [8]), or positive-semi definite imbalance (required by [2]).
> >
> > Our result in Theorem 3, however, does generalize the assumptions in [2,8,10,11] to a wider set of initialization, it improves the rate bound (Table 1), it makes connections to the general bound in Theorem 2 (Please see the remark "Connection to results for three-layer"), and it finds new effects of depth on convergence (Please see the remark "Effect of imbalance under unimodality"). While we certainly agree that unimodality will not likely hold for random (unbiased) initialization, we do think that it uncovers novel features of the learning dynamics, like the super-exponential dependence of the convergence rate on the network depth.
> >
> > If this is "looking under the lamppost", we would argue then that most existing published work on this topic falls under this category. However, it is our belief that this is the path to knowledge, that is, doing research is like making new "lampposts" until the road to the goal (understanding deep learning) is fully lighted.
> >
> >
> >
> > **Comparison with prior work**: We thank the reviewer for the references.
> >
> > [3] and other landscape-based analyses require certain landscape properties on the loss function that cannot be satisfied by even a deep linear network, and often can only provide a local convergence rate around the equilibrium. We have added a paragraph discussing [3] and similar work in the introduction (highlighted in blue).
> >
> > For deep linear networks, [4] and other kernel regime results study a different type of initialization (random initialization + large hidden layer width) than ours (unimodal weights), hence there is no substantial overlap. For three-layer networks, our bound covers random initialization. We expect that our bound would be not as good as the one from NTK analysis, but note that the strength of our result lies in its breadth (See "Bound in Theorem 2"): while NTK analysis provides a better bound, it works exclusively for the kernel regime initialization.
> >
> > For [5] and other work on the linear residual network, we feel it is impossible to have a head-to-head comparison as the network architecture is different, and so are the training dynamics. Nonetheless, we note that [5],[6] initialize all the weights by zero, which is very restrictive, and we did mention in the original draft the connection between such zero initialization on the residual net [6] and the unimodal initialization on the fully connected net, we highlight the text in blue, please see the remark "Comparison with prior work" after Theorem 2.
> >
> > We hope these clarifications are helpful. Please feel free to post more comments and questions during the discussion period. Thank you!
> >
> > References:
> >
> > [1] Hancheng Min, et al. "Convergence and implicit bias of gradient flow on overparametrized linear networks." arXiv preprint arXiv:2105.06351, 2022.
> >
> > [2] Yun, Chulhee, et al. "A unifying view on implicit bias in training linear neural networks." International Conference on Learning Representations. 2020.
> >
> > [3] Jin, Chi, et al. "How to escape saddle points efficiently." International Conference on Machine Learning. PMLR, 2017.
> >
> > [4] Hu, Wei, et al. "Provable Benefit of Orthogonal Initialization in Optimizing Deep Linear Networks." International Conference on Learning Representations. 2019.
> >
> > [5] Zou, Difan, et al. "On the Global Convergence of Training Deep Linear ResNets." International Conference on Learning Representations. 2019.
> >
> > [6] Lei Wu, et al. "Global convergence of gradient descent for deep linear residual networks." NeurIPS, 2019
> >
> > [7] Simon Du and Wei Hu. "Width provably matters in optimization for deep linear neural networks." ICML, 2019.
> >
> > [8] Sanjeev Arora, et al. "A convergence analysis of gradient descent for deep linear neural networks." ICLR, 2018.
> >
> > [9] Dominik Stöger and Mahdi Soltanolkotabi. "Small random initialization is akin to spectral learning: Optimization and generalization guarantees for overparameterized low-rank matrix reconstruction." NeurIPS, 2021.
> >
> > [10] Andrew M Saxe, et al. "Exact solutions to the nonlinear dynamics of learning in deep linear neural network." ICLR, 2014.
> >
> > [11] Suriya Gunasekar, et al. "Implicit bias of gradient descent on linear convolutional networks." NeurIPS, 2018

---

### Decision · Program_Chairs · 2023-01-20

**Decision:**

Reject

**Justification For Why Not Higher Score:**

Please see the above.

**Justification For Why Not Lower Score:**

N/A


**Metareview: Summary, Strengths And Weaknesses:**

Some early papers on the analysis of gradient flow to learn linear neural networks showed that if initialization was “balanced” in a sense, this would be maintained throughout training, and used this property to prove convergence.  An interesting recent paper presented a more refined analysis in the case of two-layer networks, that demonstrated *imbalance* can also be advantageous.  This paper extends that work, to more loss functions, and to deeper networks.

The consensus view was that the formulation of the two-layer result for a wider variety of losses was interesting, but that the technical novelty required for generalizing the analysis was somewhat limited.

The bound provided in the three-layer case was difficult to understand.  In particular, its strength was hard to judge, and some doubts about it persisted through the discussion period, which included a videoconference meeting to discuss this paper.  The reviewers were not convinced that the condition analyzed in the case of networks of depth greater than three was a significant step forward beyond a condition analyzed in previous work.


**Summary Of Ac-Reviewer Meeting:**

We went over the bounds for the three layer case and the deep case in detail, along with the new section written by the authors providing conditions under which the three-layer bound was not vacuous.  We tried to get an intuitive sense of the quality of the bounds, and what was required of an initialization to make them good.